# Conformational dynamics of cohesin/Scc2 loading complex are regulated by Smc3 acetylation and ATP binding

Aditi Kaushik [1], Thane Than[2], Naomi J. Petela[3], Menelaos Voulgaris[3], Charlotte Percival [2], Peter Daniels [2], John B. Rafferty [2], Kim A. Nasmyth[3] & Bin Hu [1] ✉

The ring-shaped cohesin complex is a key player in sister chromatid cohesion, DNA repair, and gene transcription. The loading of cohesin to chromosomes requires the loader Scc2 and is regulated by ATP. This process is hindered by Smc3 acetylation. However, the molecular mechanism underlying this inhibition remains mysterious. Here, using *Saccharomyces cerevisiae* as a model system, we identify a novel configuration of Scc2 with pre-engaged cohesin and reveal dynamic conformations of the cohesin/Scc2 complex in the loading reaction. We demonstrate that Smc3 acetylation blocks the association of Scc2 with pre-engaged cohesin by impairing the interaction of Scc2 with Smc3's head. Lastly, we show that ATP binding induces the cohesin/Scc2 complex to clamp DNA by promoting the interaction between Scc2 and Smc3 coiled coil. Our results illuminate a dynamic reconfiguration of the cohesin/Scc2 complex during loading and indicate how Smc3 acetylation and ATP regulate this process.

Sister chromatid cohesion is mediated by cohesin, a ring-shaped complex consisting of Smc1, Smc3 and the α-kleisin subunit Scc1. Smc1 and Smc3 dimerise through 'hinge' interaction and their ATPase heads are bridged by Scc1 to form a tripartite ring, which topologically entraps sister DNAs[1]. Cohesin has also been demonstrated to act as a motor and regulate chromatin structure through loop extrusion[2].

Loading cohesin to DNA requires the Scc2/4 loading complex, which catalyses ATP-dependent DNA entrapment[3,4]. The first step of loading is thought to involve Scc2 forming a transient complex with cohesin, called the cohesin/Scc2 loading complex. Subsequently, Scc2 stimulates ATP hydrolysis and opens the pre-assembled cohesin ring, permitting DNA entrapment[5,6]. Genetic and biochemical evidence suggested that this reaction might trigger the opening of the Smc1/Smc3 hinge interface[7,8]. Alternatively, recent studies proposed the Smc3/kleisin interface as the entry gate[9,10]. The loading process is more complicated than expected because cohesin exists in two different configurations. Cohesin's coiled coils are normally juxtaposed (J-cohesin), restricting head engagement[11]. ATP binding drives head engagement (E-cohesin) and the separation of head-end coiled coils, which are crucial for ATP hydrolysis and loading (Fig. 1A). Recent Cryo-EM studies revealed that Scc2 can interact with both J- and E-cohesin[10,12–14]. In Scc2/E-cohesin, Scc2 interacts with Smc3's head and coiled coil to form a DNA clamp compartment, reflecting an essential step during loading. In addition, other Scc2/cohesin configurations are found during loop extrusion[15], although their roles in loading are unclear. These discoveries implied a dynamic reconfiguration of the Scc2/cohesin complex during the loading reaction (Fig. 1A).

DNA-associated cohesin can be released by a regulatory subunit Wapl[16–20], which induces the opening of the Smc3 coiled-coil/N-Scc1 interface, thereby permitting entrapped DNAs to escape from the cohesin ring[21]. This Wapl-dependent release poses an inherent risk to cohesion and, therefore, must be neutralised once sister chromatid cohesion is established. During DNA replication, Eco1, an acetyltransferase transferring acetyl groups to Smc3 K112 and K113[16,22,23], antagonises the Wapl's releasing activity[21]. Surprisingly, pre-acetylated cohesin fails to establish cohesion and a deacetylase, Hos1 in yeast or

[1]The Institute of Medical Sciences, University of Aberdeen, Aberdeen AB25 2ZD, UK. [2]Department of Molecular Biology and Biotechnology, University of Sheffield, Sheffield S10 2TN, UK. [3]Department of Biochemistry, University of Oxford, Oxford OX1 3QU, UK. ✉e-mail: bin.hu@abdn.ac.uk

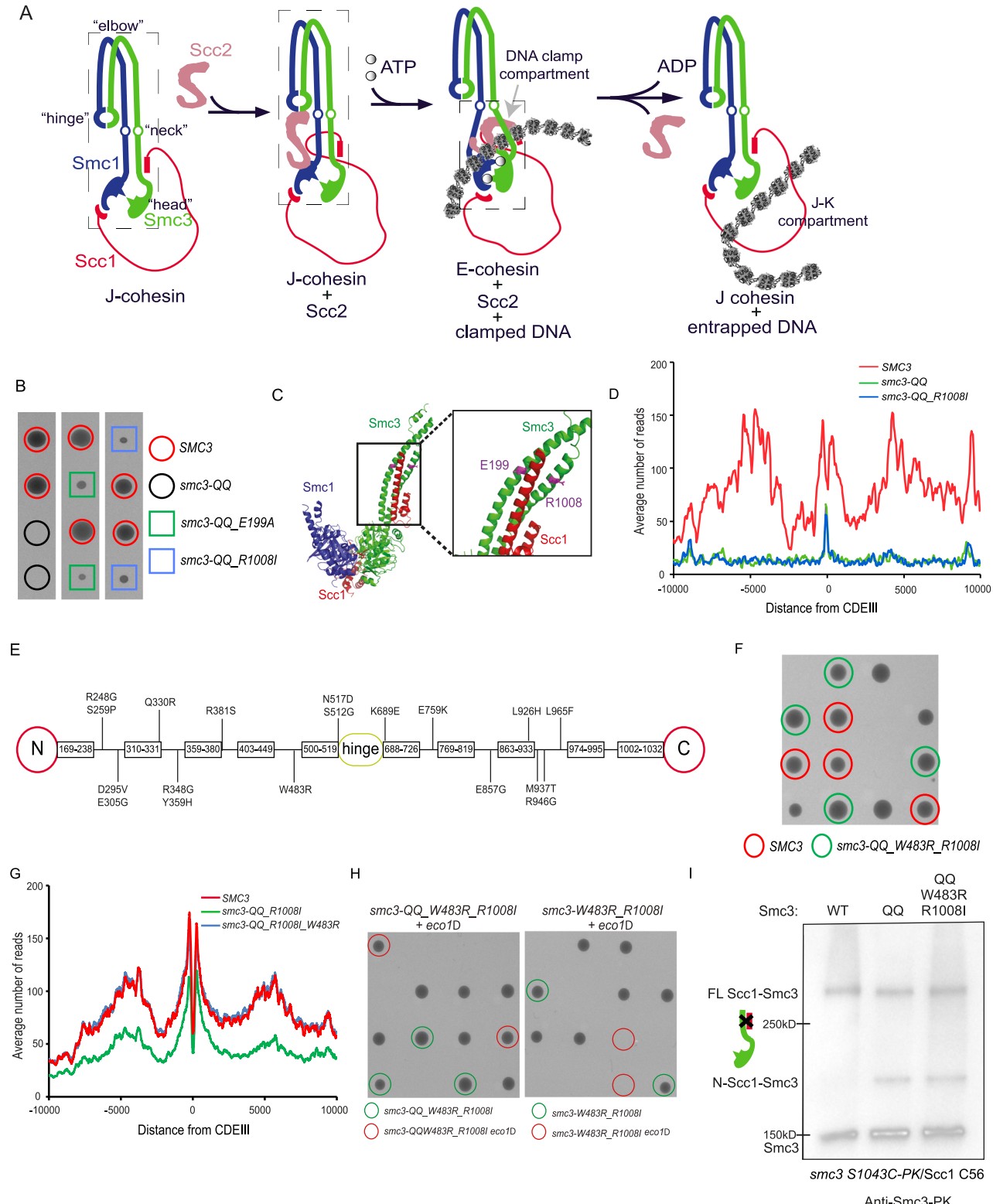

HDAC8 in humans, is required to recycle acetylated cohesin by removing acetyl groups[24–26]. A clue to how acetylated Smc3 hinders cohesion derives from a study of the acetylation mimicking form of *SMC3*, *smc3-K112Q K113Q* (referred to simply as *smc3-QQ* hereinafter)[27]. Similar to pre-acetylated cohesin, this version of Smc3 cannot create cohesion. Calibrated ChIP-seq revealed that *smc3-QQ* significantly impairs cohesin loading in vivo[27]. This implies that Smc3 acetylation not only counteracts Wapl's releasing activity but also inhibits the loading reaction. It is not clear how this modification blocks these

opposite processes. One possibility is that Smc3 acetylation simply prevents the opening of the Smc3/kleisin interface because this opening is claimed to be required by both loading and release[10,21]. Alternatively, Smc3 acetylation might regulate these processes via different mechanisms. To understand the role of Smc3 acetylation in loading and release, these possibilities must be clarified.

Here, we investigated how Smc3 acetylation inhibits the loading reaction using the acetylation mimicking mutant *smc3-QQ*. We first demonstrated that Smc3 acetylation impacts loading and release

**Fig. 1 | Smc3 acetylation affects loading and release through distinct mechanisms. A** Summary of recent discoveries of cohesin loading: cohesin is normally presented with juxtaposed coiled coils (J-cohesin). The Scc2 forms a complex with J-cohesin through Scc2/Smc1 head interaction. Then, Scc2/cohesin complex is recruited to DNA. In this configuration, cohesin heads are engaged by ATP binding (E-cohesin), Scc2 interacts with both heads, and DNA is clamped in a compartment enclosed by Scc2 and Smc3 coiled coil. After ATP hydrolysis, E-cohesin is transformed into J-cohesin, and DNA is entrapped in the J-K compartment. The squares with dash lines indicate the structures revealed by Cryo-EM analysis (EMD-12880 and PDB 6ZZ6). **B** Tetrad dissection. *smc3 R008I* and *smc3 E199A* partially restored cell proliferation of *smc3 QQ* at 25 °C. Source data is provided as a Source Data file. **C** Positions of Smc3 E199 and R1008 on Smc3's coiled coil based on Smc3-Scc1 crystal structure (PDB 4UX3). **D** Calibrated ChIP-seq was performed in the exponential cells of *SMC3, smc3-QQ* and *smc3-QQ_R1008I*. DNA association of Smc3$^{QQ}$ and Smc3$^{QQ\_R1008I}$ is inhibited. Source data is provided as a Source Data file. **E** The distributions of additional suppressor mutations for R1008I on Smc3 coiled coil (**F**) Tetrad dissection. *R1008I_W483R* restored the growth of *smc3-QQ*. The growth of *SMC3* (red circles) and *smc3-QQ_W483R_R1008I* (green circles) cells are indistinguishable. Source data is provided as a Source Data file. **G** Calibrated ChIP-seq profiles of exponentially grown *SMC3, smc3-QQ_R1008I*, and *smc3-QQ_W483R_R1008I*. *smc3 W483R_R1008I* greatly improved the DNA association of Smc3$^{QQ}$. Source data is provided as a Source Data file. **H** Tetrad dissection. *smc3-QQ_W483R_R1008I*, but not *smc3-W483R_R1008I* can bypass the requirement of Eco1 for cell growth. Source data is provided as a Source Data file. **I** Exponentially grown yeast cells were subject to in vivo BMOE crosslink. The non-crosslinked Smc3 and Smc3 crosslinked to N-Scc1 and full-length Scc1 in whole cell extract were analysed by western blot using anti-PK antibody. Source data is provided as a Source Data file.

through independent pathways. We then identified a novel transitional configuration in addition to the two reported configurations and established the dynamics of conformational changes in the Scc2/cohesin complex during loading. We further showed that Smc3 acetylation inhibits the transitional configuration by blocking the interaction of Scc2 with the Smc3 head. Additionally, we revealed that ATP binding promotes Scc2/Smc3 coiled-coil interaction to regulate DNA clamping. In light of our results, we propose a dynamic reconfiguration of the Scc2/cohesin complex during loading regulated by Smc3 acetylation and ATP binding.

## Results

### Screening of mutations suppressing the lethality of *smc3-QQ*

To understand the roles of Smc3 acetylation in loading and release, we identified two suppressor mutations *smc3 E199A* and *smc3 R1008I*, which partially restored the proliferation of *smc3-QQ* cells (Fig. 1B). Though at opposite ends of the Smc3 polypeptide, E199 and R1008 are situated close to each other on opposite strands of Smc3's anti-parallel coiled coil, in a part that forms a three helical bundle with the α3 helix within Scc1's NTD[28] (Fig. 1C). This suppressive effect was fairly weak as the growth of *smc3-QQ R1008I* cells was poor at 25 °C and stopped at 37 °C (Supplementary Fig. 1A). In fact, *R1008I* barely suppressed the loading defect of ectopically expressed Smc3$^{QQ}$, which might be due to the competition of endogenous Smc3 (Fig. 1D). To enable further characterisation of the suppressing effect, we screened for intragenic mutations using random mutagenesis and identified 34 mutations that permit the growth of *smc3-QQ R1008I* at 37 °C. We identified 18 suppressor mutations distributed across both strands of Smc3 coiled coil (from R248 to N517 and from K689 to L965), revealing a crucial role of coiled coil in loading (Fig. 1E). Without exception, all of these mutations require the primary mutation *R1008I* to permit the proliferation of *smc3-QQ* cells, implying a central role of the R1008 region in loading. In this study, we used suppressor mutations, *smc3 W483R_R1008I*, to investigate the role of Smc3 acetylation in loading, because the growth of *SMC3* and *smc3-QQ_W483R_R1008I* cells are almost indistinguishable (Fig. 1F).

### *smc3 W483R_R1008I* greatly suppresses *smc3-QQ*'s defect in loading but not release

To measure the DNA association of Smc3$^{QQ\_W483R\_R1008I}$ without interference from endogenous Smc3, we replaced *SMC3* with *smc3-QQ_R1008I* or *smc3-QQ_W483R_R1008I*. Calibrated ChIP-seq showed modest DNA association of Smc3$^{QQ\_R1008I}$, consistent with the ability of *smc3-QQ_R1008I* to weakly support cell growth (Fig. 1G). Strikingly, *W483R_R1008I* greatly improved the loading of Smc3$^{QQ}$ (Fig. 1G).

We next addressed whether these suppressor mutations also restored the releasing activity of Smc3$^{QQ}$. Inhibition of releasing activity by Eco1 is critical for cohesion and cell viability[16,22,23], therefore, the recovered releasing activity of Smc3$^{QQ\_W483R\_R1008I}$ will lead to lethality in *eco1Δ* cells. Tetrad analysis showed that *smc3-QQ_W483R_R1008I*

bypassed the requirement of Eco1 for cell growth (Fig. 1H). This effect is due to the acetylation-mimicking mutations, *K112Q K113Q* because Eco1 is still essential in *smc3-W483R_R1008I cells* (Fig. 1H). This genetic evidence demonstrated that *smc3-QQ* can inhibit the releasing activity. Importantly, the loading suppressor mutation *W483R_R1008I* does not release this inhibition. This is confirmed by directly evaluating the disassociation of the separase-cleaved N-terminal Scc1 fragment from Smc3$^{QQ\_W483R\_R1008I}$, which requires Wapl's releasing activity, as previously described[29]. In this assay, a cysteine pair *smc3 S1043C* and Scc1 C56 on the Smc3 coiled-coil/N-Scc1 interface was crosslinked in vivo using bismaleimidoethane (BMOE), a homobifunctional chemical for covalently crosslinking two sulfhydryl groups within 6-11 Å[30]. As previously reported, the release of N-Scc1 from Smc3 is greatly impaired by *smc3-QQ*, and this defect cannot be remedied by *W483R_R1008I* (Fig. 1I).

These results revealed that *smc3-QQ* is not just a structural analogue of acetylation but it also functionally mimics Smc3 acetylation to inhibit Wapl's releasing activity and maintain cohesion. This allows us to use *smc3-QQ* to investigate the biochemical properties of Smc3 acetylation. Moreover, *W483R_R1008I* can fully suppress the loading defect of Smc3$^{QQ}$, but not its release defect, revealing that Smc3 acetylation prevents loading and release through different mechanisms. Hereby, we focused on studying its role in loading.

### *smc3-QQ* inhibits Scc2-stimulated ATP hydrolysis of cohesin

To understand how Smc3 acetylation inhibits loading, we first evaluated the effect of *smc3-QQ* on the Scc2-stimulated ATPase activity of cohesin. In this assay, a recombinant tetramer of cohesin (Smc1, Smc3 or Smc3$^{QQ}$, Scc1, Scc3) was incubated with ATP and GFP-Scc2[4]. Consistent with previous reports, ATPase activity of the wild-type tetramer was detected only upon stimulation by Scc2 and DNA[3,4] (Fig. 2A). However, Scc2's ability as a stimulator of ATPase activity was significantly undermined by *smc3-QQ* when DNA is absent (Fig. 2B). This suggests that *smc3 QQ* either impair the intrinsic ATPase activity of cohesin or inhibit the Scc2/cohesin interaction required for triggering ATP hydrolysis. Surprisingly, DNA greatly recovered the Scc2-dependent ATP activity of Smc3$^{QQ}$, although not to a WT level (Fig. 2B and Supplementary Fig. 2A). This result suggests that DNA might facilitate ATP hydrolysis-compatible conformation of cohesin, for example, head engagement, or stabilise cohesin/Scc2 interaction, even when Smc3 is acetylated. Although Smc3 K112 K113 was proposed as a DNA sensor to promote the DNA/cohesin association[9], our results suggested that it might not be the main function of these residues in loading.

### *smc3-QQ* barely affects ATP-mediated head engagement and Scc2/cohesin complex assembly

As Smc3 K112 and K113 are adjacent to the ATP binding pocket, their acetylation might alter its conformation and inhibit head engagement[28]. To survey this, we examined ATP-mediated head

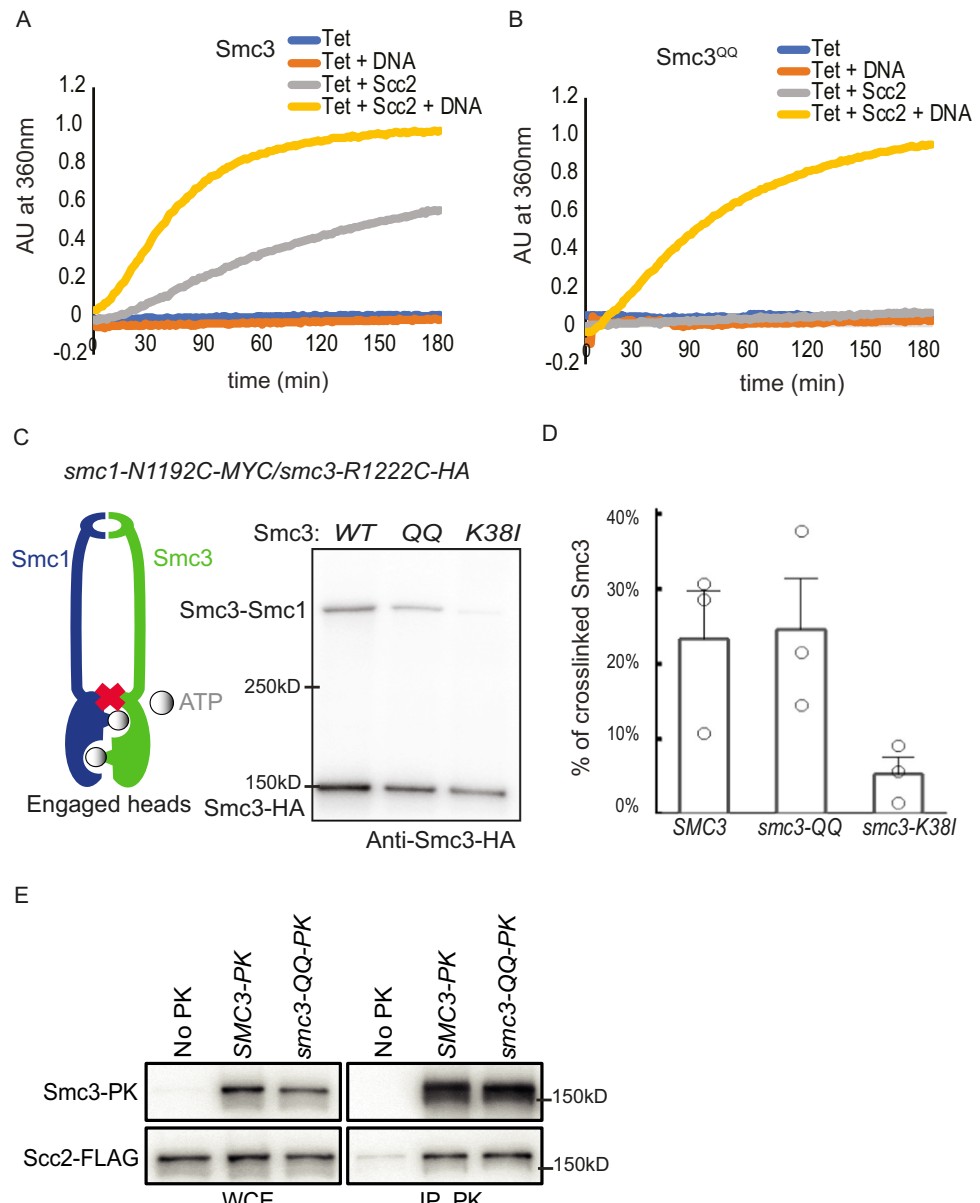

**Fig. 2 | *smc3 QQ* affects cohesin's interaction with Scc2 rather than DNA.**
**A** Recombinant tetramer of cohesin (Smc1, Smc3, Scc1 and Scc3) was incubated with DNA or Scc2 or both. ATP was added to initiate the reaction, and the reaction rate was measured as the change in absorption at 360 nm over time. Source data is provided as a Source Data file. **B** Recombinant tetramer of cohesin (Smc1, Smc3QQ, Scc1 and Scc3) was incubated with DNA or Scc2 or both. ATP was added to initiate the reaction, and the reaction rate was measured as the change in absorption at 360 nm over time. Source data is provided as a Source Data file. **C** In vivo BMOE crosslink between *smc1 N1192C* and *smc3 R1222C* in the exponentially grown cells. representing the head engagement between Smc1 and Smc3, in the presence of

*SMC3, smc3-QQ* or *smc3-K38I*. Cohesin complexes were immunoprecipitated using anti-PK antibody for Scc1-PK. Crosslink bands were separated by 3–8% gradient gel and non-crosslinked and crosslinked Smc3 was examined by Western Blot using anti-HA antibody. Source data is provided as a Source Data file. **D** The percentage of crosslinking efficiency in (**C**) was calculated as mean+SD from three independent experiments. WT crosslinking efficiency was 23%, Smc3QQ was 24% and Smc3K38I was reduced to 5%. Source data is provided as a Source Data file. **E** Co-immunoprecipitation of Scc2 with Smc3 or Smc3QQ was carried out with exponentially grown cells. No difference in co-immunoprecipitating Scc2 between Smc3 or Smc3QQ was recorded. Source data is provided as a Source Data file.

engagement using the cysteine pair, *smc1 N1192C* and *smc3 R1222C*, which can be crosslinked by bismaleimidoethane (BMOE) only when the head engagement is triggered by ATP[11]. In this experiment, the exponentially grown cells were incubated with 2.5 mM BMOE. After lysis, the cohesin complex was immunoprecipitated using an anti-PK antibody against Scc1-PK and Smc3 or Smc1 was examined by Western blot (Fig. 2C and Supplementary Fig. 2B). Approximately 23% of Smc3[R1222C] formed a covalent crosslink product with Smc1[N1192C] (Fig. 2D). This crosslink occurs in an ATP-dependent manner because the ATP-binding mutant *smc3-K38I*[31] significantly reduced the crosslinking efficiency (5%) (Fig. 2D). Interestingly, *smc3-QQ* has little effect

on the crosslink, with a crosslink efficiency of 24%. A similar conclusion was drawn by examining the Smc1[N1192C] using anti-Myc antibody (Supplementary Fig. 2C). This result excluded the possibility that the defective loading of acetylated Smc3 is due to impaired head engagement.

Next, we examined whether Smc3 acetylation might prevent the Scc2/cohesin interaction. Co-immunoprecipitation revealed that Smc3[QQ] displayed a comparable ability to interact with Scc2 as wild-type Smc3 (Fig. 2E). This indicates that the global interaction between Scc2 and cohesin was barely impaired by Smc3 acetylation.

## Scc2 interacts both Smc1 and Smc3 head domains

Although *smc3-QQ* has little effect on the assembly of the Scc2/cohesin complex, This mutant might alter a local contact and disrupt an essential Scc2/cohesin configuration. Based on this rationale, we performed an in vivo BPA crosslink screen to identify Scc2/cohesin interfaces. Considering its role in ATP hydrolysis, we thus reasoned three following regions of the Smc3 head as potential interaction sites: (1) Smc3 acetylation site-KKD strand, consisting of two conserved lysines (K112 and K113) and one aspartic acid residues (D114); (2) the adjacent α helix (R58-L64), presumably communicating between the KKD strand and the Smc3 ATP-binding pocket[28]; (3) the regions surrounding the Smc3 ATP-binding pocket. A given residue (K57, R58, R61, H66, Q67, M74, L111, or Q117) in these regions was substituted with p-benzoyl-l-phenylalanine (BPA) (Fig. 3A), a photoreactive amino acid that can crosslink to any other residues within 7 Å upon UV irradiation[32,33]. We did not detect any Smc3-Scc2 crosslink at the KKD strand (L111BPA and Q117BPA) (Fig. 3B), not as predicted by the in vitro Scc2/E-cohesin model[13]. This might be because this interaction would be highly sensitive to ATP-mediated head engagement, which is transient in WT cohesin[8]. Interestingly, we found that Scc2 was crosslinked with *smc3 K57BPA, Q67BPA,* or *M74BPA,* among which *smc3 Q67BPA* showed the highest crosslinking efficiency and formed two distinct crosslink products (Fig. 3B and Supplementary Fig. 3A). Increasing the size of Scc2 with a 6xFlag tag led to an upshift of the top crosslink band, as expected for an Smc3-Scc2 crosslink (Supplementary Fig. 3A). This was confirmed by Western blot using an anti-FLAG antibody. This result reveals a physical interaction between Scc2 and the Smc3 ATP-binding domain (Fig. 3A). Given that head engagement brings Smc3 Q67 close to Smc1's head domain (Fig. 3A), *smc3 Q67BPA* was crosslinked to Smc1, as expected (Supplementary Fig. 3B).

We also carried out a BPA screen in a region of Smc1 near the ATP-binding pocket (Fig. 3C) and identified three adjacent BPA substitutions, *smc1 T1100BPA, E1102BPA, and K1113BPA*, which can crosslink to Scc2 (Fig. 3D). *smc1 E1102BPA* also crosslink to Smc3 (Supplementary Fig. 3C). To our surprise, Smc1-Scc2 crosslinks were barely observed for *smc1 F1123BPA, E1127BPA,* and *Y1128BPA*, since head engagement brings these residues close to Smc3 Q67 (Fig. 3C). Hence, this result implies that heads might be not engaged when Scc2 interacts with Smc3 Q67. Through these experiments, we found Scc2 interacts with two regions (Smc3 Q67 and Smc1 E1102) around ATP binding pocket on Smc1 and Smc3 heads.

## Both interfaces co-exist in the same Scc2/cohesin configuration

To define the interface between Scc2 and Smc3 Q67, we developed a novel strategy to map Scc2 sites crosslinked by *smc3 Q67BPA*. We created a set of functional Scc2 alleles, each containing 3x TEV protease cleavage sequences at the indicated sites (Supplementary Fig. 3D). All these versions of Scc2 are labelled with a 6xFLAG tag at the C-terminus, except Scc2[TEV1176], because the FLAG-fusion destroys its function. The Smc3-Scc2 crosslink products were subjected to TEV protease digestion, producing an untagged Scc2 N-terminal fragment and a FLAG-tagged C-terminal fragment (Supplementary Fig. 3D). Western blot showed that *smc3 Q67BPA* crosslinked to the C-terminal fragment up to residue 1109, while it crosslinked to the N-terminal fragments when cleaved at positions 1176 and 1222 (Supplementary Fig. 3D). This revealed that *smc3 Q67BPA* crosslinked to Scc2 1110-1176.

To further refine this interface, we replaced 15 residues across this region and Smc3 Q67 with cysteine (Supplementary Fig. 3E). Interestingly, six cysteine-substituted alleles of *scc2 (R1115C, N1133C, E1158C, D1162C, E1168C,* and *R1173C)* are synthetically lethal with *smc3-Q67C,* implying an essential role of this interface in loading. Of the remaining nine cysteines, *scc2 S1171C* or *T1175C* showed the highest BMOE crosslink efficiency with *smc3 Q67C* revealed by in vivo BMOE crosslink, indicating the interface between Smc3 Q67 and Scc2 S1171/T1175 (Figs. S3E and 3E). Using the same strategy, we found that *smc1*

*E1102BPA* crosslinked to Scc2 1260-1303 (Supplementary Fig. 3F) and *smc1 E1102C* efficiently crosslinked to *scc2 T1281C* (Fig. 3F and Supplementary Fig. 3G).

To test whether both interfaces would exist in one Scc2/cohesin configuration or belong to two different configurations, we introduced both cysteine pairs into endogenous Smc1/Smc3/Scc2 and examined whether these two crosslinks could occur simultaneously. Indeed, the Scc2/cohesin complex bearing both cysteine pairs generated not only two single crosslink products but also an Smc1-Smc3-Scc2 double crosslink product (Fig. 3G). The crosslink strictly took place between the cysteine pair of *smc3 Q67C/scc2 T1175C* or *smc1 E1102C/scc2 T1181C* since interchanged cysteine pairs cannot be crosslinked (Fig. 3G). The double crosslink demonstrates that the two interfaces (*smc3 Q67C/scc2 T1175C* and *smc1 E1102C/scc2 T1281C*) co-exist in the same Scc2/cohesin configuration (Fig. 3G). Based on the proximity of these two cysteine pairs, we constructed a structural model of this configuration (Fig. 3H).

## Smc3 Q67/Scc2 interface represents a novel Scc2/cohesin configuration

In both reported configurations of Scc2/J-cohesin[14] and Scc2/E-cohesin[10,12,13], the predicted distances between Smc3 Q67 and Scc2 T1175 are about 16 Å, beyond the BMOE-crosslink distance (6–11 Å)[30] (Supplementary Fig. 4A, B), suggesting that Smc3 Q67/Scc2 interface might represent a novel cohesin/Scc2 configuration. However, we cannot exclude the possibility that this crosslink could still occur in the Scc2/E-cohesin complex due to the flexible Q67 loop. Interestingly, a residue Smc3 M74 in this loop is found to be in the vicinity of Scc2 in both our modelled Scc2/cohesin and Scc2/E-cohesin configurations, consistent with our result that *smc3 M74BPA* can crosslink to Scc2 (Fig. 3B). However, Smc3 M74 is predicted next to different Scc2 residues, Scc2 N886 in our modelled structure and E821 in Scc2/E-cohesin configuration (Supplementary Fig. 4C, D), which is confirmed by BMOE crosslink (Fig. 4B and Supplementary Fig. 4E). If these crosslinks take place only in the Scc2/E-cohesin configurations due to the flexibility of the Q67/M74 loop, allowing Smc3 M74 to oscillate between Scc2 N886 and E821, this movement will also bring Smc3 M74 close to Scc2 R830, which locates in-between these two residues (Fig. 4A). However, this possibility was omitted as no crosslinking was observed between the corresponding cysteine pair (*smc3 M74 C/scc2 R830C*) (Fig. 4B and Supplementary Fig. 4E), consistent with our claim of a novel Scc2/cohesin configuration indicated by the *smc3 Q67C/scc2 T1175C* crosslink.

The structural modelling predicted two features of the new configuration: open coiled coils and disengaged heads. Neither E nor J-cohesin fulfils both criteria, implying that Scc2 would not crosslink to *smc3 Q67BPA* in either J-cohesin or E-cohesin. To test this, we first chemically stabilised J-cohesin or E-cohesin by crosslinking the juxtaposed coiled coils or engaged heads using BMOE. We then examined whether *smc3 Q67BPA* can crosslink to Scc2 (Fig. 4C, D; left panels). If BMOE-stabilised J-cohesin or E-cohesin still interacts with Scc2, the following BPA crosslink would produce a triple crosslink product among Smc1-Smc3-Scc2. Otherwise, the BMOE crosslink of Smc1/Smc3 and BPA-crosslink of *smc3 Q67BPA*/Scc2 would be mutually exclusive. In this experiment, *smc1 E1102BPA* was used as a positive control, as the Smc1 E1102/Scc2 interface presents in both Scc2/J-cohesin and Scc2/E-cohesin configurations[10,12–14]. A unique feature of J-cohesin is the juxtaposed head-end coiled coils, as shown in our modelled structure (Supplementary Fig. 4F). Extensive contact predicted by this model was examined by in vivo BMOE crosslink of 7 cysteine pairs along the coiled coils (Supplementary Fig. 4G). The pair (*smc1 R1031C/smc3 E202C*) with the highest crosslink efficiency was used to crosslink coiled coils and thus covalently fix the J-cohesin. As predicted, *smc1 E1102BPA*, but not *smc3 Q67BPA*, was able to crosslink with Scc2 when the juxtaposed coiled coils were stabilised by a BMOE

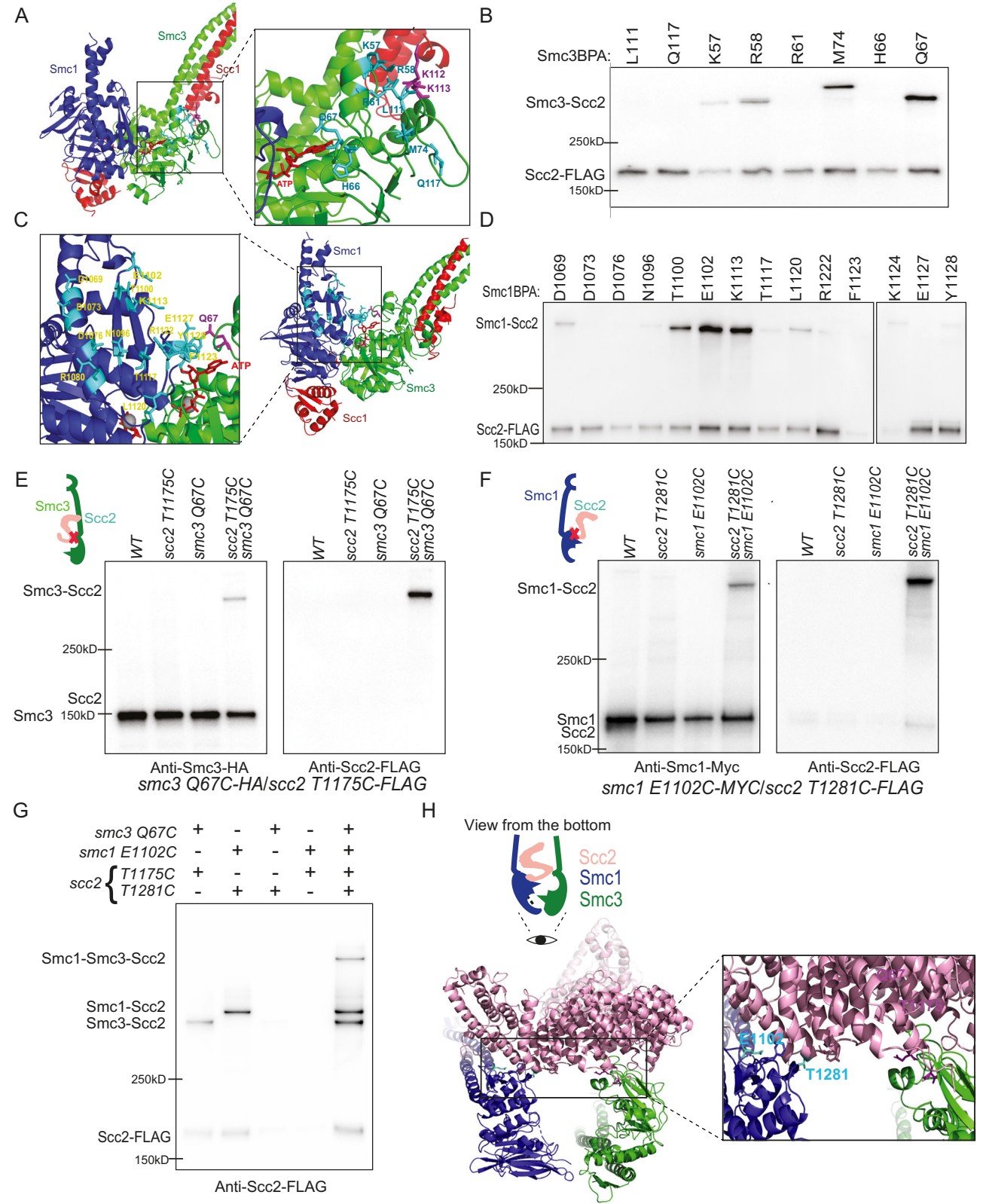

crosslink (Fig. 4C and Supplementary Fig. 4H). We repeated this experiment with engaged heads crosslinked using a cysteine pair (*smc1 N1192C* and *smc3 R1222C*)[11]. Unlike *smc1 E1102BPA*, *smc3 Q67BPA* did not crosslink with Scc2 when heads were engaged (Fig. 4D, Supplementary Fig. 4I, J). These results show that the cohesin/Scc2 configuration characterised by the Smc3 Q67/Scc2 interface is distinct from the Scc2/J-cohesin or Scc2/E-cohesin configurations, indicating that

Smc3 Q67/Scc2 interface represents a novel configuration, which we named as Scc2/pre-E-cohesin.

### ATP-dependent head engagement promotes DNA clamping
The structural modelling of the Scc2/pre-E-cohesin complex (Fig. 3H) predicts that this configuration, unlike Scc2/E-cohesin, cannot form ATP binding pocket and therefore is incompatible with ATP-mediated

**Fig. 3 | Mapping of Smc1-Scc2 and Smc3-Scc2 interfaces. A** The indicated residues near the Smc3 KKD strand and ATP binding pocket based on the crystal structure of Smc3-Scc1 (PDB 4UX3) were substituted by BPA. **B** In vivo BPA crosslinking of selected Smc3 residues. After UV crosslink, Scc1-PK was immunoprecipitated from whole-cell extracts and co-immunoprecipitated crosslinked and non-crosslinked Scc2 were detected by western blot using an anti-FLAG antibody. *smc3 K57BPA, R58BPA, Q67BPA* and *M74BPA* successfully crosslinked with Scc2, in which *smc3 Q67BPA* showed better crosslink efficiency than the rest three. Source data is provided as a Source Data file. **C** The indicated residues around the Smc1 ATP binding pocket based on the crystal structure of Smc1-Scc3 (PDB 1W1W) were substituted by BPA. **D** BPA crosslink for selected Smc1 residues. After in vivo UV crosslink, Scc1-PK was immunoprecipitated from whole-cell extracts and co-immunoprecipitated crosslinked and non-crosslinked Scc2 were detected by western blot using an anti-FLAG antibody. *smc1 E1102BPA* and *smc1 K1113BPA* showed higher crosslink efficiency. All the samples derive from the same experiment and both Western blots were processed in parallel. Source data is provided as a Source Data file. **E** In vivo BMOE crosslinking revealed close proximity between Smc3 Q67 and Scc2 T1175. Single or both residues were replaced by cysteine and in vivo BMOE crosslinking was performed using exponentially grown cells. Scc1-PK was immunoprecipitated from whole-cell extracts and co-immunoprecipitated crosslinked and non-crosslinked Smc3 and Scc2 were detected by western blot using anti-HA and anti-FLAG antibodies. Successful crosslink product formation depended on the presence of both cysteines. Source data is provided as a Source Data file. **F** In vivo crosslinking revealed close proximity between Smc1 E1102 and Scc2 T1281. Single or both residues were replaced by cysteine and in vivo BMOE crosslinking was performed using exponentially grown cells. Scc1-PK was immunoprecipitated from whole-cell extracts and co-immunoprecipitated crosslinked and non-crosslinked Smc3 and Scc2 were detected by western blot using anti-HA and anti-FLAG antibodies. Successful crosslink product formation depended on the presence of both cysteines. Source data is provided as a Source Data file. **G** Simultaneous in vivo BMOE crosslinks of *smc3 Q67C/scc2 T1175C* and *smc1 E1102C/scc2 T1281C* were performed in exponentially grown cells. Cohesin complexes were immunoprecipitated using anti-PK antibody for Scc1-PK separated on gradient gel, and Scc2 was analysed by Western Blot using anti-FLAG antibody. Source data are provided as a Source Data file. **H** Structural model of Scc2-cohesin configuration reported in this study, showing the positions of Smc3 Q67/Scc2 T1175 and Smc1 E1102/Scc2 T1281.

head engagement. To test this, we decided to check the effects of ATP hydrolysis mutants on formation of the three different configurations, J, E and pre-E cohesin. For this, we need to establish a method to examine the Scc2/E-cohesin configuration in vivo. As suggested by Cryo-EM studies[10,12,13], this configuration is maintained by three interfaces: Smc1 E1102/Scc2 T1281, and two unique interfaces, Scc2/Smc3 head and Scc2/Smc3 coiled coil—forming a compartment for DNA entrapment (Fig. 5A). To detect these interfaces in vivo, we introduced cysteine pairs *smc3 S72C/scc2 E819C* into Smc3 head/Scc2 or *smc3 K1004C/scc2 D369C* into Smc3 coiled-coil/Scc2 interfaces (Fig. 5A). Both cysteine pairs can be specifically crosslinked in vivo using BMOE, verifying these interfaces of the Scc2/E-cohesin configuration in vivo (Fig. 5B, C, Supplementary Fig. 5A, B). We next examined whether the Scc2/J-cohesin, pre-E-cohesin, and E-cohesin conformations would be affected by ATP hydrolysis mutants *smc3 E1155Q* or *smc1 E1158Q*, which blocks ATP hydrolysis and is likely to increase E-cohesin by slowing down its transition to J-cohesin[8]. The ATP hydrolysis mutants barely affected the Smc-Scc2 crosslink in Scc2/J-cohesin (*smc1 E1102C/scc2 T181C*) and Scc2/pre-E-cohesin (*smc3 Q67C/scc2 T1175C*) (Fig. 5D, Supplementary Fig. 5C, D). These results are not surprising because, in these two configurations, the heads are disengaged and cannot form a proper ATP binding pocket. The ATP hydrolysis mutant *smc3 E1155Q* improves the Scc2/E-cohesin configuration. It doubled E-Smc3 head/Scc2 crosslink (*smc3 S72C/scc2 E819C*) and increased E-Smc3 coiled-coil/Scc2 crosslink (*smc3 K1004C /scc2 D369C*) by about 10 times (Fig. 5D, Supplementary Fig. 5C, D). These results suggested that ATP binding and head engagement would orient Smc3 coiled coil and promote the interaction between Smc3 coiled coil with Scc2. Because this interface is critical for the DNA-clamp compartment formation[10,12,13], hence, head engagement should play a major role in forming this compartment and subsequent DNA entrapment. To examine the importance of head engagement in Scc2/E-cohesin compartment in vivo, both cysteine pairs on the interfaces of E-Smc3 head/Scc2 (*smc3 S72C/scc2 E819C*) and E-Smc3 coiled-coil/Scc2 (*smc3 K1004C/scc2 D369C*) were introduced into Scc2 and Smc3/Smc3^E1155Q. As expected, the crosslink of E-Smc3 coiled-coil/Scc2 (*smc3 K1004C/scc2 D369C*) is inefficient with wild-type *SMC3* and dramatically increased by the ATP hydrolysis mutation *smc3 E1155Q* to a similar level of the E-Smc3 head/Scc2 (*smc3 S72C/scc2 E819C*) (lane 4, Fig. 5E). An extra crosslink band appeared at the top of western blot for *smc3-E1155Q* cells, indicating simultaneous crosslink of both interfaces and representing the DNA-clamp compartment, as seen in the Scc2/E-cohesin cryo-EM structure[10,12,13]. The formation of this compartment strictly depends on ATP-mediated head engagement since the double crosslink product is barely detectable in wild-type *SMC3* cells (lane 1, Fig. 5E).

To examine if this compartment would also be present in the Scc2/pre-E-cohesin complex, we repeated double crosslinks of pre-E-Smc3 head/Scc2 (*smc3 Q67C/scc2 T1175C*) and E-Smc3 coiled-coil/Scc2 (*smc3 K1004C/scc2 D369C*). The double crosslink band in the Scc2/pre-E-cohesin complex was much weaker, about 15% of those in the Scc2/E-cohesin complex (Fig. 5F). Unlike 1:1 stoichiometric crosslink of E-Smc3 head/Scc2 (*smc3 S72C/scc2 E819C*) and E-Smc3 coiled-coil/Scc2 (*smc3 K1004C/scc2 D369C*) (lane 5, Fig. 5F), the crosslink efficiency of pre-E-Smc3 head/Scc2 (*smc3 Q67C/scc2 T1175C*) is double in comparison to E-Smc3 coiled-coil/Scc2 (*smc3 K1004C/scc2 D369C*) (lane 4, Fig. 5F). This proved that the crosslinks between of pre-E-Smc3 head/Scc2 (*smc3 Q67C/scc2 T1175C*) and E-Smc3 coiled-coil/Scc2 (*smc3 K1004C/scc2 D369C*) are mutually exclusive and hence, Smc3 Q67/Scc2 T1175 and Smc3 K1004/Scc2 D369 interfaces represent two different configurations. The weak double crosslink in the Scc2/pre-E-cohesin complex might indicate a configurational transition from Scc2/pre-E-cohesin to Scc2/E-cohesin. All these data uncover one of the essential roles of ATP-dependent head engagement, which modulates the orientation of the Smc3 coiled coil and promotes its interaction with Scc2.

### *smc3-QQ* impairs the interaction of Scc2 with pre-E- or E-cohesin, but potentially not J-cohesin

Identification of the Scc2/pre-E-cohesin configurations, together with two reported configurations, allows us to elucidate the effect of Smc3 acetylation on loading by investigating how *smc3-QQ* affects these configurations. We first surveyed the effect of *smc3-QQ* on the Scc2/pre-E-cohesin configuration by examining the *smc3 Q67C/scc2 T1175C* crosslink. To avoid the interference caused by Smc3 acetylation, cells were arrested in the late G1 phase by overexpressing a nondegradable Sic1(9 m)[21], which inhibits S phase-CDK and prevents DNA replication and Smc3 acetylation. The crosslink of Scc2/pre-E-cohesin (*smc3 Q67C/scc2 T1175C*) is dramatically reduced by *smc3-QQ*, to about 10% of wild-type Smc3 (Fig. 6A, B and Supplementary Fig. 6A). However, *smc3-QQ* might only distort this interface and lead to the misalignment of these two residues. In this situation, Smc3 Q67 might still interact with other Scc2 residues. To this end, we evaluated the effect of *smc3-QQ* on *smc3 Q67BPA*/Scc2 crosslinks. The acetylation-mimicking mutant scarcely affects the crosslink of *smc3 Q67BPA* with Smc1 (Supplementary Fig. 6B), consistent with our claim that Smc3 acetylation has little effect on head engagement (Fig. 2D). However, the *smc3-QQ Q67BPA*/Scc2 crosslink product was barely detectable (Supplementary Fig. 6B), indicating that *smc3-QQ* seriously impairs the Scc2/pre-E-cohesin configuration. Since Pds5 and Scc2 compete for cohesin binding[4], Smc3^QQ might preferentially interact with Pds5, inhibiting the formation of the Scc2/pre-E-cohesin complex. This possibility was omitted

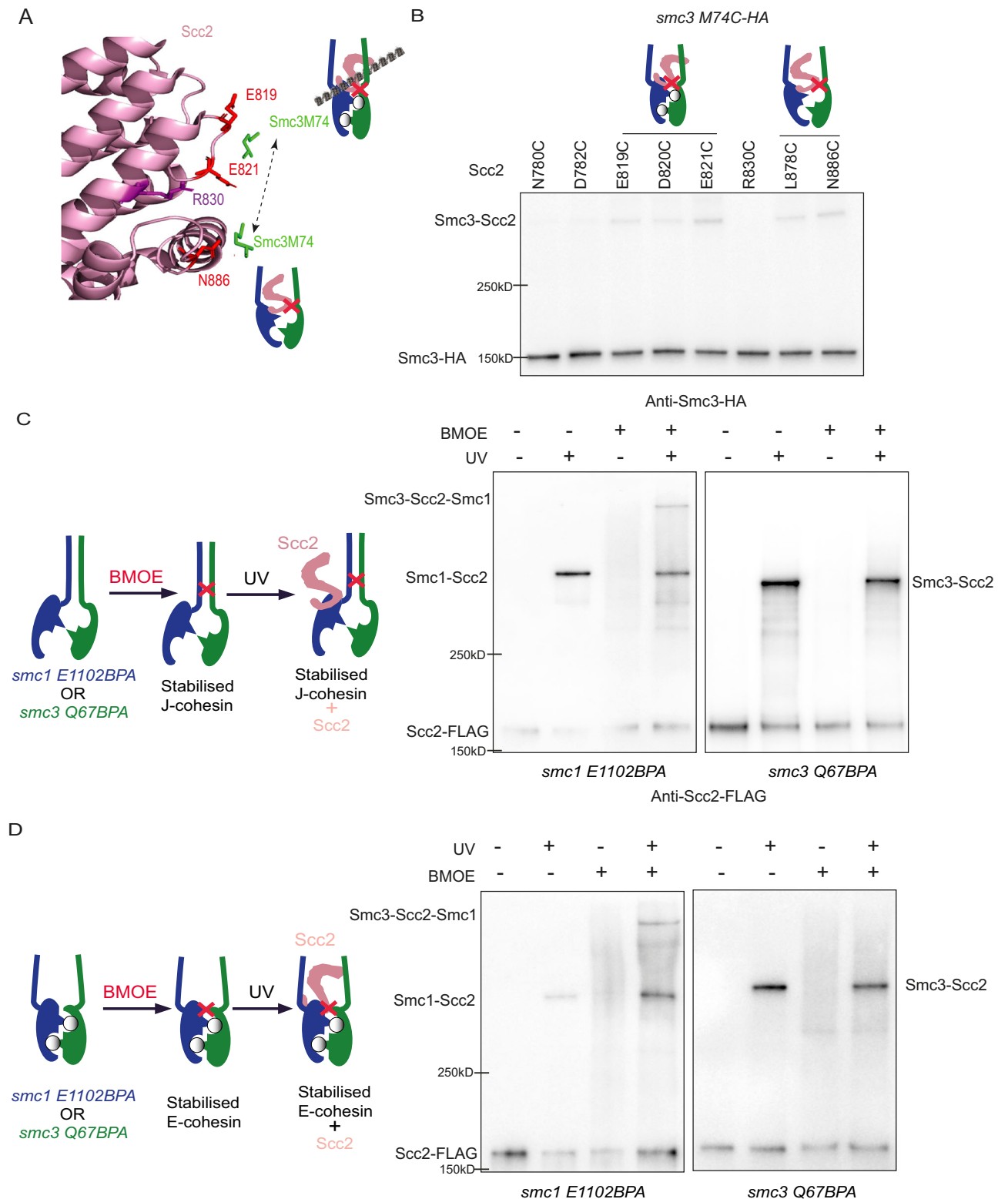

since Pds5 depletion using auxin-inducible degradation did not improve the *smc3-QQ Q67BPA*/Scc2 crosslink (Supplementary Fig. 6C).

We next asked if *smc3-QQ* could affect the Scc2/E-cohesin. To examine that, we used the two unique interfaces of E-Smc3 head/Scc2 (*smc3 S72C/scc2 E819C*) and E-Smc3 coiled-coil/Scc2 (*smc3 K1004C/scc2 D369C*). We found that *smc3-QQ* dramatically decreased the crosslinks of both interfaces by about 90%, compared to the wild type (Fig. 6E, F, Supplementary Fig. 5D, E), which is similar to the

inhibitory effect of *smc3-QQ* on the Scc2/pre-E-cohesin represented by the *smc3 Q67C/scc2 T1175C* crosslink. These results suggested that *smc3-QQ* impairs both the Scc2/pre-E-cohesin and Scc2/E-cohesin configurations.

Even though *smc3-QQ* dramatically impairs the Scc2/pre-E-cohesin and Scc2/E-cohesin configurations, as co-IP revealed that Smc3$^{QQ}$ can still interact with Scc2 (Fig. 2E), we deduced that the Scc2/J-cohesin configuration might be not affected by *smc3-QQ*. However, we could

**Fig. 4 | Smc3 Q67/Scc2 T1175 interface represents a novel Scc2/cohesin conformation. A** The relative positions of Smc3 M74 with Scc2 E819, E820, R830, and N886 in Scc2/E- (PDB 6ZZ6) or pre-E-cohesin configuration. **B** The in vivo BMOE crosslink of *smc3 M74C* with indicated cysteine substituted Scc2 in exponentially grown cells. Scc1-PK was immunoprecipitated from whole-cell extracts and co-immunoprecipitated crosslinked, and non-crosslinked Smc3-HA was detected by western blot using an anti-HA antibody. Source data is provided as a Source Data file. **C** In vivo BPA crosslink of *smc1 E1102BPA* (left) or *smc3 Q67BPA* (right) to Scc2 in J-cohesin using exponentially grown cells. Juxtaposed coiled coils were stabilised by in vivo BMOE crosslink between *smc1 R1031C* and *smc3 E202C*, followed by UV-induced BPA crosslink of *smc1 E1102BPA* or *smc3 Q67BPA* with Scc2. Scc1-PK was immunoprecipitated from whole-cell extracts and co-immunoprecipitated crosslinked and non-crosslinked Scc2-FLAG was detected by Western blot using anti-FLAG antibody. **D** In vivo BPA crosslink of *smc1 E1102BPA* (left) or *smc3 Q67BPA* (right) to Scc2 in E-cohesin using exponentially grown cells. Engaged heads were stabilised by in vivo BMOE crosslink between *smc1 N1192C* and *smc3 R1222C*, followed by UV-induced BPA crosslink of *smc1 E1102BPA* or *smc3 Q67BPA* with Scc2. Scc1-PK was immunoprecipitated from whole-cell extracts and co-immunoprecipitated crosslinked and non-crosslinked Scc2-FLAG was detected by Western blot using anti-FLAG antibody. Source data is provided as a Source Data file.

not identify a cysteine pair specific to this configuration because the only Scc2/J-cohesin interface (Smc1 E1102/Scc2 T1281) is also present in the other two configurations. If *smc3-QQ* would have a limited effect on the crosslink of this interface, the Scc2/J-cohesin configuration might not be affected by *smc3-QQ*, given that this mutant dramatically decreased the Smc/Scc2 crosslink in the other two configurations. To test this, we ectopically expressed Smc3-PK or Smc3$^{QQ}$-PK in cells bearing endogenous *smc1-E1102C/scc2-T1281C*. The crosslinked Smc1-Scc2 associated with Smc3-PK or Smc3$^{QQ}$-PK complex was immunoprecipitated using an anti-PK antibody and examined by Western blot (Fig. 6G). In contrast to the pre-E-cohesin (*smc3 Q67C*) or E-cohesin (*smc3 S72C*)/Scc2 crosslinks, the *smc1 E1102C*/Scc2 crosslink was mildly affected by *smc3-QQ* (Fig. 6G, H). This implied that Smc3 acetylation might not affect the Scc2/J-cohesin configuration. It is not unexpected because Scc2/Smc1 head interface (Scc2 T1281/Smc1 E1102) in this configuration is not close to Smc3 K112 K113. However, we cannot exclude the possibility that this interface miught be also present in other undiscovered Scc2/cohesin configurations which are not affected by Smc3 acetylation.

### *smc3-W483R_R1008I* improves Scc2's interaction with Smc3$^{QQ}$'s head and coiled coil

We have demonstrated that the suppressor mutation on Smc3 coiled coil, *smc3-W483R_R1008I*, greatly remedies the loading defect of Smc3$^{QQ}$ (Fig. 1F); thus this mutation probably improves the configurations impaired by *smc3-QQ*. To examine this, we evaluated the effect of *W483R_R1008I* on the Scc2/pre-E-cohesin crosslink in Smc3$^{QQ}$ (represented by *smc3-QQ Q67C/scc2 T1175C*). We found that *R1008I* mildly improved this crosslink (about 40% of wild-type Smc3 with Scc2), but *W483R* has little effect (Fig. 7A, B and Supplementary Fig. 7A). The combination of *R1008I* and *W483R* greatly increased the crosslinking efficiency, about 70% of the wild type. This was confirmed using *smc3 QQ Q67BPA*/Scc2 crosslink (Supplementary Fig. 7B). Similar effects were also found on the Scc2/E-Smc3$^{QQ}$ head interface (*smc3 S72C/scc2 E819C*) (Supplementary Fig. 7C, D). These results suggested that configurational modification of Smc3 coiled coil conferred by *W483R_R1008I* might tune Smc3$^{QQ}$'s head, facilitating its interaction with Scc2. Interestingly, *R1008I*, regardless of *W483R*, improved Scc2/E-Smc3 coiled-coil interaction indicated by *smc3 K1004C/scc2 D369C* crosslinks, about 70% of the wild type (Fig. 7C, D and Supplementary Fig. 7E). This suggests that another role of *R1008I* is to improve the interaction of Scc2 with Smc3 coiled coil, which is supported by, R1008's proximity to the Scc2/E-Smc3 coiled-coil interface.

We therefore suspected that the defective Scc2-dependent ATPase activity of Smc3$^{QQ}$ can also be improved by *R1008I W483R*. However, *W483R_R1008I* mildly increased the Scc2-dependent ATPase activity of Smc3$^{QQ}$ in the absence of DNA (Fig. 7E), although it greatly improved the DNA association of Smc3$^{QQ}$ (Fig. 1G). It revealed that the main role of these suppressor mutations is to improve the interaction of Smc3$^{QQ}$ with Scc2 and form the Scc2/pre-E-cohesin complex. This permits its DNA association which further enhances ATP hydrolysis. This explains why *W483R_R1008I* can greatly improve the loading of Smc3$^{QQ}$, although

mildly increased Scc2-dependent ATPase activity in the absence of DNA.

### Smc3 acetylation might impair cohesin's intrinsic ATPase activity

The fact that *smc3-W483R_R1008I* greatly improved the Scc2/Smc3$^{QQ}$ head interaction but not the Scc2-dependent ATPase activity raises a possibility that the intrinsic ATPase activity might be affected by *smc3-QQ* or Smc3 acetylation. Consistently, we identified 16 suppressors of *smc3-QQ_R1008I*, which mutations locate on the Smc3 head regions responsible for ATP binding/hydrolysis, including Q-loop, signature motif, and ATPase binding pocket (Supplementary Fig. 7F). Thus, we proposed that enhancing ATP hydrolysis would remedy the growth defect of *smc3-QQ*. To test this, we examined the effect of *scc2-E822K_L937F (scc2-EKLF)*, a hypermorphic Scc2 allele[4], on the growth of *smc3-QQ* or *smc3-QQ_R1008I* mutant. Strikingly, although *scc2-EKLF* cannot suppress the lethality of *smc3-QQ*, it greatly promotes cell proliferation of *smc3-QQ_R1008I* mutant (Fig. 7F). Further tetrad dissection revealed that either *scc2-E822K* or *L937F* is sufficient to improve the growth of *smc3-QQ_R1008I* cells (Supplementary Fig. 7G). We ectopically expressed Smc3, Smc3$^{QQ}$, or Smc3$^{QQ_R1008I}$ in wild-type *SCC2* or *scc2-EKLF* cells and examined their DNA association using calibrated ChIP-seq (Fig. 7G). Although *scc2-EKLF* barely affected the loading of Smc3$^{QQ}$, it significantly elevated the DNA association of Smc3$^{QQ_R1008I}$ from centromere to chromosomal arms. Intriguingly, the ATP hydrolysis of Smc3$^{QQ}$ and Smc3$^{QQ_R1008I}$ in the absence of DNA were dramatically increased by *scc2-EKLF* (Fig. 6H, I). These results implied that the hypermorphic Scc2 allele can compensate for the impaired intrinsic ATPase activity of *smc3-QQ*. However, this compensation is insufficient for its loading and still requires *smc3-R1008I*, although *R1008I* barely contributes to the ATPase activity of Smc3 or Smc3$^{QQ}$ (Fig. 7I and Supplementary Fig. 7H). Considering the role of *smc3-R1008I* in the interaction of Scc2 with Smc3$^{QQ}$ coiled coil, we inferred that this mutation would facilitate the interaction of Scc2$^{EKLF}$ with Smc3$^{QQ}$ coiled coil to form the DNA-clamp compartment.

## Discussion

Cohesin loader Scc2/4 complex is essential for its various functions, such as sister chromatid cohesion and loop extrusion[34–36]. Recently Cryo-EM studies revealed that Scc2 can bind both J- and E-cohesin complexes[10,12–14]. However, it is unclear what are their roles in loading and how these configurations are regulated. In this study, we confirmed both configurations in vivo. Interestingly, we identify another Scc2/cohesin configuration (Scc2/pre-E-cohesin) (Fig. 3H), defined by two interfaces between Scc2 and Smc1/Smc3 heads. The interface between Scc2 and Smc3 head indicated by Scc2 T1175/Smc3 Q67 is unique to this configuration, while Scc2/Smc1 head interface (Scc2 T1281/Smc1 E1102) is also present in two other two configurations. Several lines of evidence demonstrated that this Scc2/pre-E-cohesin configuration is distinguishable from the other two. First, *smc3 Q67BPA* is mainly crosslinked to Scc2$^{1110–1176}$ in this configuration (Supplementary Fig. 3D). But in the Scc2/E-cohesin configuration, Smc3 Q67 is much closer to Scc2 K1278 (7.7 Å) and Y1279 (9.5 Å) than T1175 (16.5 Å).

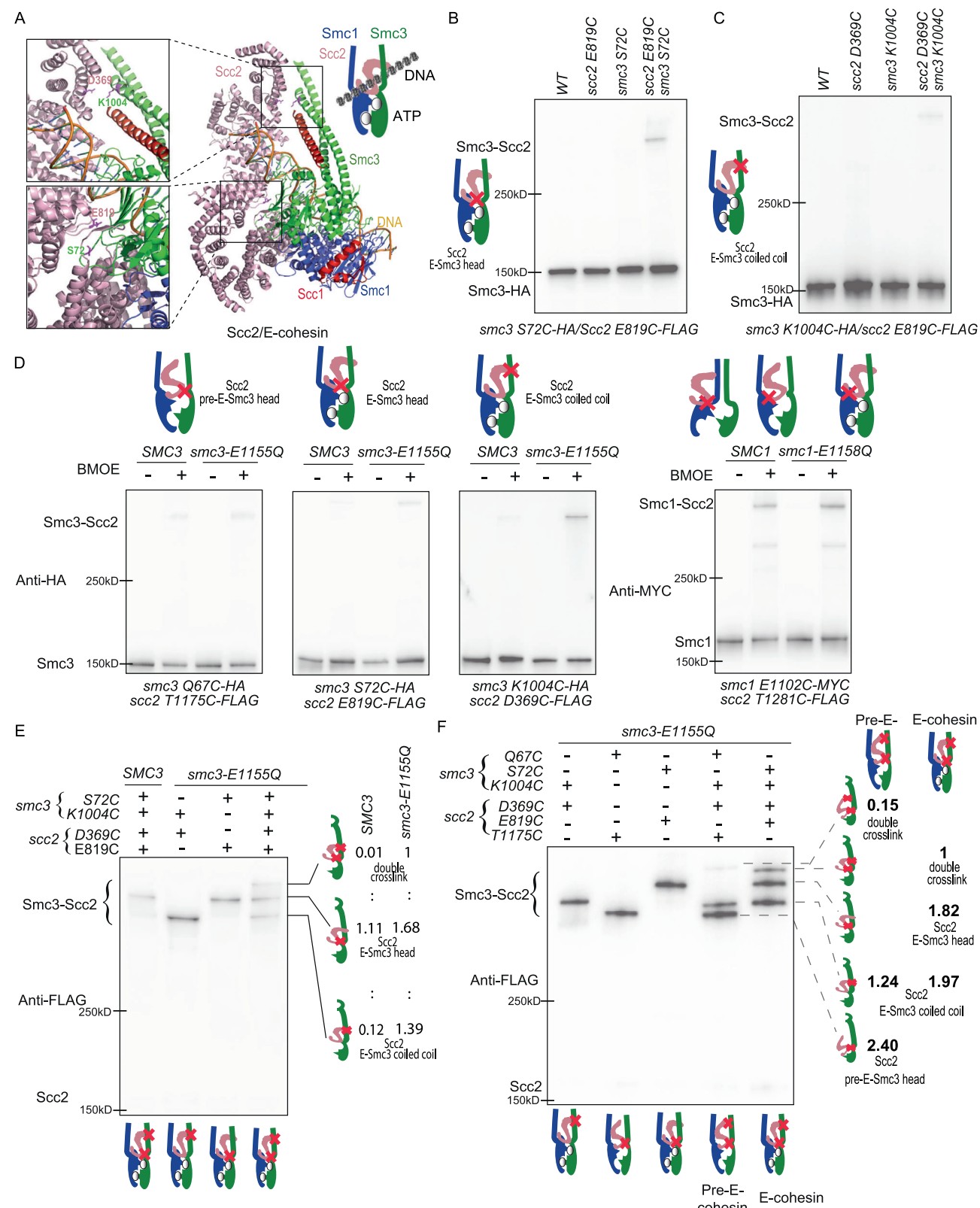

This suggested that Scc2 T1175/Smc3 Q67 interface represents a different configuration (Scc2/pre-E-cohesin). The privileged crosslink of *smc3 Q67BPA* with Scc2[1110–1176] implied that the Scc2/pre-E-cohesin configuration might predominate over Scc2/E-cohesin. This is likely because ATP hydrolysis triggers the disassembly of the Scc2/E-cohesin complex, making Scc2/E-cohesin less stable. Second, this configuration is incompatible with J-cohesin because juxtaposed coiled coils

impose a structural restraint on the Scc2/Smc3 head interaction in the Scc2/pre-E-cohesin configuration (Fig. 4C). Using the same strategy, we demonstrated that head engagement also prevents the assembly of this interface (Fig. 4D). These results are consistent with the prediction from our modelled structure: the Scc2/pre-E-cohesin complex displays separated coiled coils and disengaged heads. Another piece of evidence is that the interfaces of Scc2/pre-E-cohesin head (Scc2 T1175/

**Fig. 5 | Inhibition of Scc2/pre-E-cohesin and Scc2/pre-E-cohesin configurations by *smc3-QQ*. A** Crystal structure of E-cohesin/Scc2 complex showing the positions of Smc3 and Scc2 residue pairs smc3 S72/Scc2 E819 and Smc3 K1004/Scc2 D369. (PDB 6ZZ6). **B** In vivo BMOE crosslinking between *smc3 S72C/scc2 E819C* in exponentially grown cycling cells. Scc1-PK was immunoprecipitated from whole-cell extract, co-immunoprecipitated crosslinked, and non-crosslinked Smc3-HA was detected by western blot using an anti-HA antibody. Successful crosslink product formation depended on the presence of both cysteines. Source data is provided as a Source Data file. **C** In vivo BMOE crosslinking between *smc3 K1004C/scc2 D369C* in exponentially grown cycling cells. Scc1-PK was immunoprecipitated from whole-cell extracts, co-immunoprecipitated crosslinked, and non-crosslinked Smc3-HA was detected by western blot using an anti-HA antibody. Successful crosslink product formation depended on the presence of both cysteines. Source data is provided as a Source Data file. **D** Effect of cohesin ATP hydrolysis mutant (*smc3 E1155Q* or *smc1 E1158Q*) on the in vivo BMOE crosslinks on different Scc2/cohesin interfaces. Ectopically expressed wild type or EQ mutant versions of Smc3[S72C], Smc3[K1004C], Smc3[Q67C], or Smc1[E1102C] were crosslinked to their respective cysteine substitution of Scc2 residues using BMOE in late G1 arrested cells Scc1-PK was immunoprecipitated from whole-cell extracts and coimmunoprecipitated Smc3 or

Smc1 was detected by western blot using anti-HA or anti-MYC antibody. Crosslinking efficiency was quantified using band intensity on western blot. Source data is provided as a Source Data file. **E** DNA clamp compartment formed by the interfaces of Smc3 S72/Scc2 E819 and Smc3 K1004/Scc2 D369. Simultaneous in vivo BMOE crosslink of *smc3 S72C/scc2 E819C* and *smc3 K1004C/scc2 D369C* with *SMC3* or *smc3-E1155Q* was performed using cells arrested in late G1. Scc1-PK was immunoprecipitated from whole-cell extracts and coimmunoprecipitated Scc2 was detected by western blot using an anti-FLAG antibody. The single and double crosslink products were quantified using band intensity on western blot. Their stoichiometric ratios were indicated on the right. Source data is provided as a Source Data file. **F** Double crosslink of *smc3 Q67C/scc2 T1175C* and *smc3 K1004C/scc2 D369C* in presence of *smc3 E1155Q*. Simultaneous in vivo BMOE crosslink of *smc3 Q67C/scc2 T1175C* and *smc3 K1004C/scc2 D369C* with *smc3-E1155Q* was performed, with indicated control, using cells arrested in late G1. Scc1-PK was immunoprecipitated from whole-cell extracts, and coimmunoprecipitated Scc2 was detected by western blot using an anti-FLAG antibody. The single and double crosslink products were quantified using band intensity on western blot. Their stoichiometric ratios were indicated on the right. Source data is provided as a Source Data file.

---

Smc3 Q67) and Scc2/E-cohesin coiled coil (Smc3 K1004/Scc2 D369) are mutually exclusive (Fig. 5F). These results support our claim that the interface of Scc2 T1175/Smc3 Q67 represents a novel configuration, Scc2/pre-E-cohesin. This hypothesis is further supported by the distinctive sensitivities of these three configurations to the acetylation mimicking mutation *smc3 QQ* and ATP hydrolysis mutation *smc3 E1155Q/smc1 E1158Q*. The Scc2/pre-E-cohesin and Scc2/E-cohesin, but not Scc2/J-cohesin, are seriously defected by *smc3 QQ* (Fig. 6B, D, F). It suggested that the interaction of Scc2 with J-cohesin is the first step in loading. Also, the ATP hydrolysis mutation only enhances the Scc2/E-cohesin (Fig. 5D), suggesting the occurrence of Scc2/J-cohesin and Scc2/pre-E-cohesin before the Scc2/E-cohesin. Together, these results support the notion that Scc2/pre-E-cohesin is a novel configuration and acts as a transitional step during loading (Fig. 7J).

How does the Scc2/pre-E-cohesin configuration promote the loading reaction? As suggested by a Cryo-EM study, Scc2 initiates the loading reaction by interacting with J-cohesin[14]. However, the juxtaposed coiled coils in this Scc2/J-cohesin configuration prevent head engagement and the assembly of the DNA-clamp compartment. In the Scc2/pre-E-cohesin configuration, the interactions of Scc2 with both heads brings them close, facilitating ATP binding and head engagement. Moreover, these interactions also drives the unzipping of juxtaposed coiled coils, permitting the interaction of Scc2 with Smc3 coiled coil and the formation of the DNA clamp compartment. In the Scc2/E-cohesin complex, Scc2 interacts with the ATP-bound cohesin[10,12,13]. If this configuration forms prior to cohesin/DNA association, Scc2 could trigger ATP hydrolysis and disassemble of this complex before loading. Another problem is that the engaged heads promote the interaction of Scc2 with Smc3 heads and coiled coil (Fig. 5D), which encloses the DNA-clamp compartment and prevents DNA entry. Unlike the Scc2/E-cohesin configuration, ATP binding pocket is not formed in the Scc2/pre-E-cohesin complex, preventing premature ATP hydrolysis. Moreover, in this configuration, Scc2 interacts with the Smc3 head but not the coiled coil, which creates an entry gate for DNA clampping. The subsequent ATP binding drives Scc2/Smc3 coiled-coil interaction and closes the gate. These structural features of the Scc2/pre-E-cohesin configuration permit Scc2 to load J-cohesin and form the DNA-clamped Scc2/E-cohesin configuration. Identifying the Scc2/pre-E-cohesin conformation also opens the door to an intermediate conformation in other Smc proteins like condensin, Smc5/6 where the engaged head DNA-bound conformations have been reported in cryo-EM studies[37,38]. Furthermore, the reconfiguration of the cohesin/Scc2 complex in loading might also occur in loop extrusion. It is conceivable that the cohesin/Scc2 complex would keep switching

between the Scc2/pre-E-cohesin and Scc2/E-cohesin configurations during the ATP binding/hydrolysis cycle in the "swing and clamp" process of loop extrusion[15].

Acetylation of Smc3 K112 K113 is critical to prevent DNA dissociation of cohesin by antagonising Wapl's releasing activity[16,21,22,39]. Ironically, this modification also precludes Scc2-mediated DNA association as revealed by studying its mimicking form *smc3-QQ*[9,27]. These observations suggest that Smc3 K112 K113 plays essential roles in both loading and release. The identified mutation in this study, *smc3-W483R_R1008I*, permitting the loading of *smc3-QQ*, proved that *smc3-QQ* can functionally copy Smc3 acetylation to prevent releasing activity and maintain cohesion. Moreover, this mutation largely restores the DNA association of Smc3[QQ] but has little effect on its release defect, suggesting that Smc3 K112 K113 participates in loading and release through distinctive pathways.

How do Smc3 K112 K113 promote loading? The Cryo-EM strucrure of Scc2/E-cohesin/DNA revealed that these two residues are close to DNA or two conserved negatively charged residues E821 E822 of Scc2[10,12]. Therefore, two hypotheses were proposed: positively charged K112 K113 acts as a DNA sensor by interacting with negatively charged DNA or promotes the interaction with Scc2 by forming electrostatic forces with Scc2 E821 E822. Thus, neutralisation of their charge by acetylation might weaken these interactions and inhibit loading. However, our results suggest that these interactions might not account for the essential role of K112 K113 in loading. First, DNA can significantly enhance the Scc2-dependent ATPase activity of Smc3[QQ] (Fig. 2B), suggesting that neutralising the positive charges of Smc3 K112 113 does not seriously impair cohesin/DNA interaction. On the other hand, we found that neutralising charges of Scc2 E821 E822 by *scc2 E821Q/E822Q*, unlike *smc3-QQ*, barely affects cell growth. In contrast, reversing the negative charge of Scc2 E822 leads to a gain-of-function allele of Scc2 (*scc2-E822K*), which can efficiently stimulate the ATPase activity of Smc3[QQ] and Smc3. Interestingly, this allele, together with *smc3 R1008I*, suppresses the loading defect and lethality of *smc3-QQ* (Fig. 7G and Supplementary Fig. 7H). These results indicated that the interaction of Smc3 K112 K113 with DNA or Scc2 E821 E822 might not be a major contributor to loading.

This study revealed that *smc3-QQ* dramatically decreases the interaction between Scc2 and Smc3 Q67. It suggests that the acetylation might impair the Scc2/pre-E-cohesin configuration and prevents the conformational transition from Scc2/J-cohesin to Scc2/E-cohesin. Consistently, the mutation *smc3 W483R_R1008I* which improves the interaction of Scc2 with Smc3[QQ] also greatly alleviates the loading defect of Smc3[QQ]. This suggests that the main role of Smc3 K112 K113 in loading is to facilitate the formation of Scc2/pre-E-cohesin

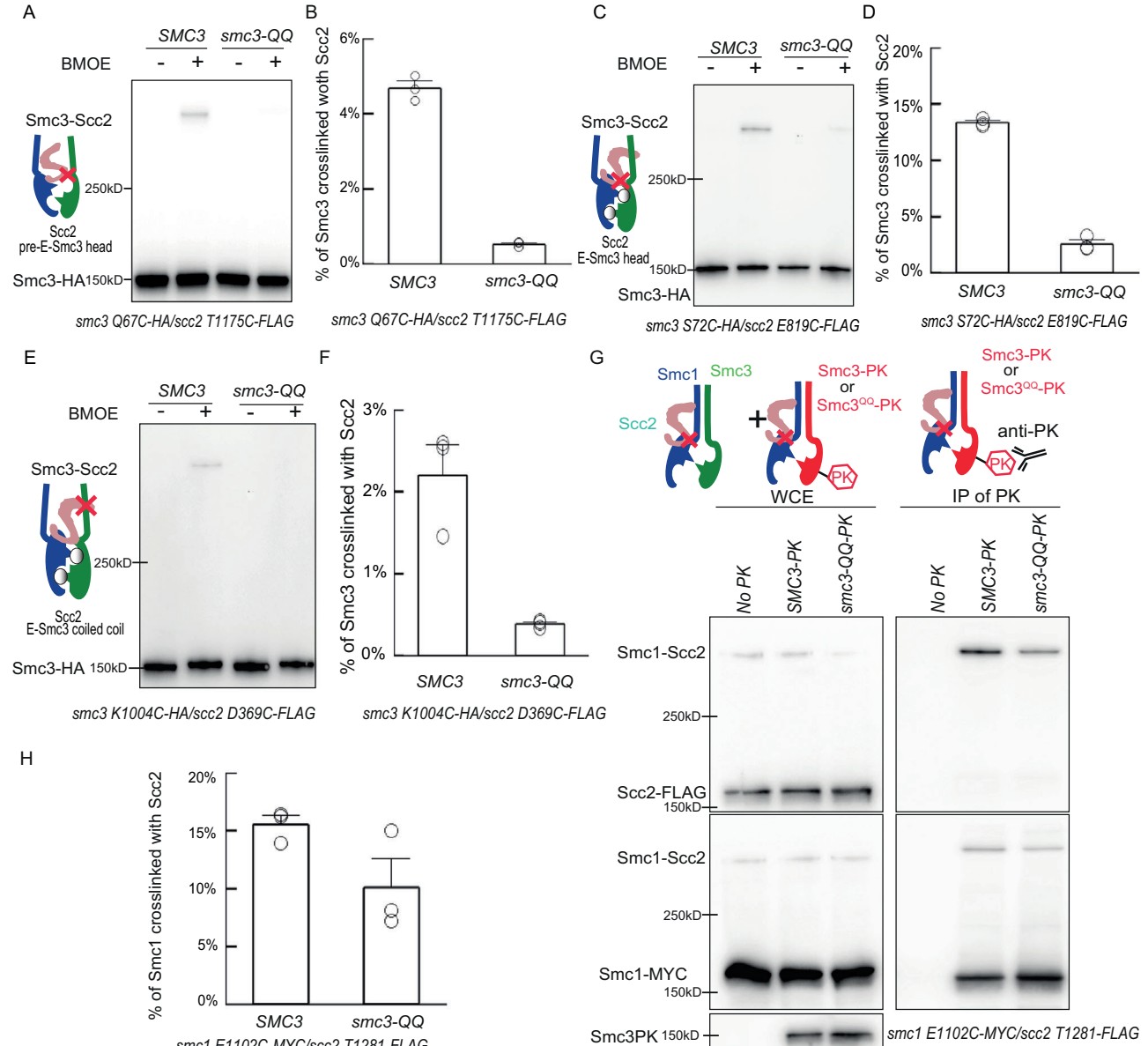

**Fig. 6 | Inhibition of Scc2/pre-E-cohesin and Scc2/pre-E-cohesin configurations by *smc3-QQ*. A** In vivo BMOE crosslink of scc2 T1175C with *SMC3* or *smc3-QQ Q67C* using cells arrested in late G1. Scc1-PK was immunoprecipitated from whole-cell extracts and co-immunoprecipitated crosslinked and non-crosslinked Smc3-HA was detected by Western blot using anti-HA antibody. Source data is provided as a Source Data file. **B** The percentage of crosslinking efficiency in panel A was calculated as mean ± SD from 3 independent experiments. *smc3-QQ* reduced the crosslinking efficiency to about 10% of wild type. Source data is provided as a Source Data file. **C** In vivo BMOE crosslinking of *scc2 E819C* with *SMC3* or *smc3-QQ S72C* using cells arrested in late G1. Scc1- PK was immunoprecipitated from whole-cell extracts and co-immunoprecipitated crosslinked and non-crosslinked Smc3-HA was detected by Western blot using anti-HA antibody. Source data is provided as a Source Data file. **D** The percentage of crosslinking efficiency in panel C was calculated as mean ± SD from 3 independent experiments. *smc3-QQ* reduced the crosslinking efficiency to about 10% of wild type. Source data is provided as a

Source Data file. **E** In vivo BMOE crosslinking of *scc2 D369C* with *SMC3* or *smc3-QQ K1004C* using cells arrested in late G1. Scc1-PK was immunoprecipitated from whole-cell extracts and co-immunoprecipitated crosslinked and non-crosslinked Smc3-HA was detected by western blot using an anti-HA antibody. Source data is provided as a Source Data file. **F** The percentage of crosslinking efficiency in panel E was calculated as mean ± SD of 3 independent experiments. *smc3QQ* reduced the crosslinking efficiency to about 10% of wild type. Source data is provided as a Source Data file. **G** In vivo BMOE crosslinking of *scc2 T1281C* with *smc1 E1102C* in the presence of ectopically expressed Smc3-PK or Smc3QQ-PK using cells arrested in late G1. Smc3/Smc3QQ-PK was immunoprecipitated from whole-cell extracts and co-immunoprecipitated crosslinked and non-crosslinked Smc1 or Scc2 was detected by western blot using anti-MYC (Smc1) and anti-FLAG (Scc2) antibody. Source data is provided as a Source Data file. **H** The percentage of crosslinking efficiency in panel J was calculated as the mean ± SD of 3 independent experiments. *smc3-QQ* had a mild effect on the crosslink. Source data is provided as a Source Data file.

configuration. However, how these residues influence this configuration still requires further investigation.

To our surprise, although the interaction of Scc2 with Smc3QQ and the loading of Smc3QQ are greatly improved by *smc3 W483R_R1008I*, this suppressor mutation has a mild effect on the reduced DNA-independent ATPase activity of Smc3QQ. This suggested that Smc3

acetylation might also impair the intrinsic ATPase activity. In agreement with this hypothesis, many mutations suppressing the growth defect of *smc3-QQ* were found in the regions involving ATP binding and hydrolysis. Moreover, we found that a hypermorphic allele of Scc2, together with *smc3 R1008I*, can efficiently suppress the loading defect of *smc3-QQ*. Consistently, a mutation in the Smc1 head greatly

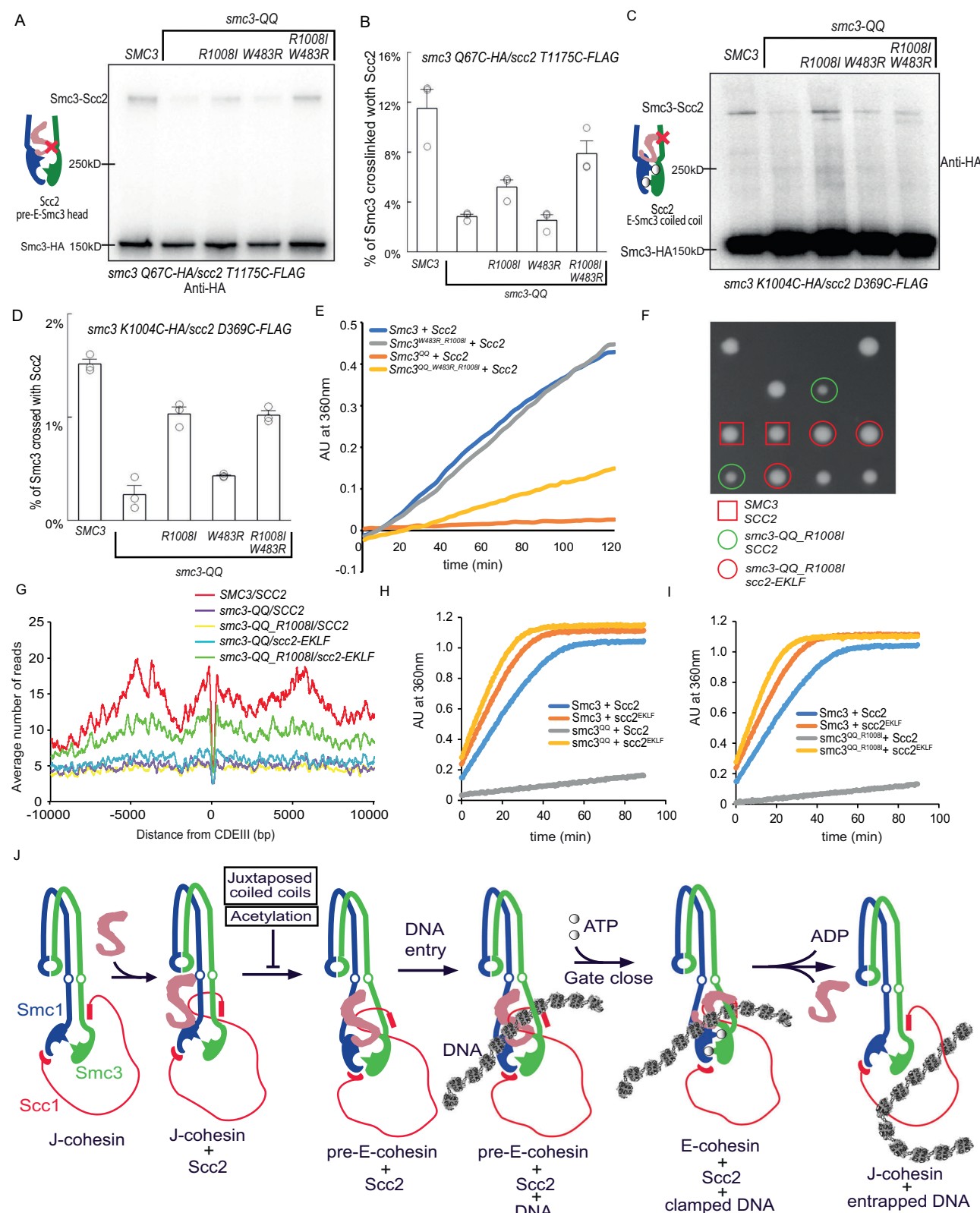

improves the intrinsic ATPase activity and suppresses the lethality of *smc3 K113Q*[40]. These results suggested that Smc3 K112 K113 would also contribute to the intrinsic ATPase activity.

Recent evidence revealed that SMC coiled coils are juxtaposed[41–43], which poses a structural barrier for Smc3/Scc2 interaction and ATP binding/hydrolysis. We demonstrated that the chemical stabilisation of juxtaposed coiled coils prevents Scc2/Smc3 Q67

interaction. Thus, the juxtaposed coiled coils must be unzipped, at least from "necks" to "heads", for assembly of the Scc2/pre-E-cohesin complex and further ATP binding/hydrolysis. Also, Scc2 interacts with Smc3 coiled coil to form a DNA clampment compartment in Scc2/E-cohesin, which also requires the unzipping of juxtaposed coiled coils. Consistently, the primary suppressor mutations *smc3 E199A* and *R1008I* localise to the coiled coil close to this interface, which might

**Fig. 7 | Loading defect of Smc3^QQ is rescued by Smc3 suppressor *R1008I* and a hypermorphic Scc2 allele. A** In vivo BMOE crosslink between *scc2 T1175C* and *smc3-QQ Q67C* in the presence of *smc3 R1008I* or *W483R* or both using cells arrested in late G1. Scc1-PK was immunoprecipitated from whole-cell extracts and co-immunoprecipitated Smc3-HA was detected by Western blot using an anti-HA antibody. Source data is provided as a Source Data file. **B** The percentage of crosslinking efficiency in panel A was calculated as mean+SD from 3 independent experiments. The highest increase in crosslink was observed when both *R1008I* and *W483R* were present, about 70% of wild type. Source data is provided as a Source Data file. **C** In vivo BMOE crosslink between *scc2 D369C* and *smc3-QQ K1004C* in the presence of *smc3 R1008I* or *W483R* or both using cells arrested in late G1. Scc1-PK was immunoprecipitated from whole-cell extracts and co-immunoprecipitated Smc3-HA was detected by Western blot using an anti-HA antibody. Source data is provided as a Source Data file. **D** The percentage of crosslinking efficiency in panel C was calculated as mean ± SD from 3 independent experiments. The highest increase in crosslink was observed when *W483R* were present, about 70% of wild type. Source data is provided as a Source Data file. **E** Recombinant tetramer of cohesin, Smc3 or Smc3^W483R_R1008I or Smc3^QQ or Smc3^QQ_R1008I_W483R, Smc1, Scc1 and Scc3 was incubated with Scc2. ATP was added to initiate the reaction and the reaction rate was measured as the change in absorption at 360 nm over time. Source data is provided as a Source Data file. **F** *scc2-E822K_L937F* (red circles)

promotes cell proliferation of *smc3-QQ R1008I* mutant (green circles), nearly to wild type (red rectangles). Source data is provided as a Source Data file. **G** Calibrated ChIP-seq profiles of Smc3, Smc3^QQ, Smc3^QQ_R1008I in the *SCC2* or *scc2-EKLF* cells. Source data is provided as a Source Data file. **H** Recombinant tetramer of cohesin, Smc3 or Smc3^QQ, Smc1, Scc1 and Scc3 was incubated with Scc2 or Scc2^E822K_L937F. ATP was added to initiate the reaction and the reaction rate was measured as the change in absorption at 360 nm over time. Source data is provided as a Source Data file. **I** Recombinant tetramer of cohesin, Smc3 or Smc3^QQ R1008I, Smc1, Scc1 and Scc3 was incubated with Scc2 or Scc2^E822K_L937F. ATP was added to initiate the reaction and the reaction rate was measured as the change in absorption at 360 nm over time. Source data is provided as a Source Data file. **J** Conformational dynamics of cohesin Scc2 loading complex. Scc2 interacts in J-cohesin with Smc1's head. Then the interaction of Scc2/Smc3 heads induces the unzipping of cohesin CC and orientates heads to facilitate the formation of ATP binding pocket (pre-E-cohesin). ATP binding leads to head engagement (E-cohesin) and promotes the assembly of the DNA clamping compartment. DNA clamping triggers ATP hydrolysis and the opening of one of the cohesin interfaces, allowing DNA to be entrapped in cohesin's J–K compartment. The transition from the configuration of Scc2/J-cohesin to that of Scc2/pre-E-cohesin is inhibited by Smc3 acetylation and juxtaposed CCs.

weaken the interaction between cohesin coiled coils and facilitate the Scc2/Smc3 coiled-coil interaction. Interestingly, this effect can be further enhanced by additional mutations occurring on the Smc3 coiled coil. The coiled coils of SMC proteins are made of several α-helix fragments of about 50 residues with conserved breaks[44]. Remarkably, most coiled-coil suppressor mutations are found to locate at joints of α-helix fragments, which might contribute to unzipping juxtaposed coiled coils by altering the orientation of the α-helix. Our investigation revealed that *smc3 W483R_R1008I* greatly improved the interaction of Scc2 with Smc3^QQ Q67. This improvement permits the assembly of the Scc2-pre-E-cohesin complex and enables the loading.

## Methods

### Reagents
Key reagents and antibodies are described in supplementary Supplementary Table 5.

### Yeast strains and growth conditions
All yeast strains were of W303 background (Supplementary Table 3). Standard yeast genetic techniques and transformation were used to construct these strains. Integration of target genes into the *MET15* locus using the CRISPR system as previously described[45]. The yeast mutations used in this study are summarised in Supplementary Table 2.

Unless otherwise stated, yeast cells were grown in the rich medium YEP + 2% glucose (YEPD) at 25 °C with a rotation of 2000 rpm (New Brunswic Innova 43R, Eppendorf).

To arrest cells in late G1 by overexpressing non-degradable Sic1(9 m)[21], exponentially grown *bar1Δ* cells in YEP + 2% raffinose (YEPR) were treated with 0.1 μg/ml α-factor. One hour before release, galactose of a final concentration of 2% was added to induce the expression of Sic1(9 m). To release cells from α-factor arrest, cells were harvested by centrifuge (3500 rpm × 2 minutes) and washed with YEPD twice. Cells were resuspended into YEPD containing 0.1 mg/ml of pronase and agitated for 60 min at 25 °C. Yeast tetrad dissection was performed using a Singer dissection microscope MSM 400.

### Screening for suppressors of scc2-45 wpl1Δ
Forty independent colonies of the parental strain (*scc2-45 wpl1Δ*) were cultured in 5 ml YEPD overnight at 25 °C. Cells collected from 1 ml culture were spread ontothree3 plates and incubated at 32 °C until colonies appeared. A colony was picked from each and confirmed robust growth at 32 °C by streaking on YEPD plate. Eighty-eight

revertants of Scc2 were identified by sequencing the Scc2 gene amplified from genomic DNA using PCR. Genetic linkage of the rest mutations with known cohesin submits (*SMC1*, *SMC3*, *SCC1*, *SCC3*, *SCC4*, *PDS5* and *ECO1*) were examined by tetrad dissection. Twenty-seven were tightly linked to SMC3 and DNA sequencing revealed numerous substitutions of just two amino acids, namely E199 and R1008.

### Screening for suppressors of *smc3-QQ_R1008I*
Random mutations were introduced into the *smc3 R1008I* gene using two-round error-prone PCR. In the first round of PCR, MnCl₂ was added to a final 30 μM to induce A/T to G/C transition. In the second round of PCR, 1 μl of the first-round PCR product was used as the template DNA, and dITP was added to a final 30μM to induce G/C to A/T transition. Both PCRs were carried out by standard PCR using G2 DNA Polymerase (Promega). The mutated smc3 R1008I and linearised YCplac111 bearing Smc3 promoter or terminator plus 50 bp sequences homologous to Smc3 ORF at each end were co-transformed into smc3-42 cells. The transformants containing suppressor mutations were selected on SD -Leu plates at 37 °C. The YCplac111 plasmids were recovered, and the mutations were identified by DNA sequencing. Sixteen suppressor mutations on the Smc3 head and twenty-four on Smc3 coiled coil were verified by replacing endogenous Smc3 with the reconstituted mutations.

### In vivo BPA crosslink
In vivo BPA crosslink was performed as previously described[46]. Yeast cells were transformed with pBH61 expressing Ec TyrRS and Ec tRNACUA, and YEplac181 containing Smc1 or Smc3 gene with a TAG mutation. These cells were grown in SD -Leu -Trp media containing 1 mM BPA at 30 °C until the OD reached 0.3-0.6 per mL. The BPA stocking solution (1 M) was made by dissolving BPA in fresh 1 N NaOH solution. The cells (40 OD) were collected, resuspended in 1 mL ice cold 1× PBS and transferred into an 8-well plate. The plate was placed on a bed of ice and put into a UV crosslinker (Spectrolinker XL-1500W crosslinker, Spectronics Corporation) equipped with a 352 nm wavelength emitting bulb (Sankyo Denki Co. Ltd). The cells were subjected to UV light at 352 nm for 3× 1 min with a 5-minute rest interval.

### In vivo BMOE crosslink
A stock solution of 125 mM BMOE and dissolved at 37 °C with shaking. First, yeast cells were exponentially grown or arrested in later G1. Then a total 30 OD of cells were collected and washed with 500 μl of PBS

twice before re-suspending in 1 ml of PBS. 25 µL of 125 mM BMOE solution dissolved in DMSO was added to the cell suspension and incubated on ice for 6 minutes. The crosslinking reaction was stopped by adding 3 µl of 1 M DTT. The cells were then washed with 1 mL of 5 mM DTT in PBS and used for protein extraction. The cysteine pairs for BOME crosslink are summarised in Supplementary Table 1.

## Co-immunoprecipitation

Proteins were extracted from 30 OD of cells in 500 µl of lysis buffer (0.05 M Tris pH 7.5, 0.15 M NaCl, 5 mM EDTA pH 8.0, 0.5% NP-40, 1 mM DTT, 1 mM PMSF, and cOmplete™ Protease Inhibitor Cocktail tablet). The crude lysis was added with 3 µl of 1 mg/ml antibody and rotated at 4 °C for 1 hour. Then 30 µl of Dynabeads (Thermo Fisher) was added and rotated at 4 °C overnight. The Dynabeads were washed three times using 1 ml of lysis buffer per wash with a 5-minute incubation with rolling at 4 °C between each wash. The lysis buffer was removed and the IPed proteins were released in 50 µL of 1× LDS sample buffer (NuPAGE® Life Technologies) by heating at 65 °C for 5 minutes.

## TEV proteinase cleavage

TEV cleavage was performed after the wash step during the immuno-precipitation. The Dynabeads were then washed once with 500 µL of digestion buffer (25 mM Tris pH 8.0, 150 mM NaCl, and 2 mM 2-Mer-captoethanol). The beads were then incubated for 16 hours at 20 °C with shaking in 30 µL of digestion buffer supplemented with 10 µl of TEV protease (Sigma). The samples were mixed with 10 µl of 1× LDS sample buffer and heated at 65 °C for 5 minutes.

## Protein gel electrophoresis and Western blotting

The crosslinked samples in 1X LDS sample buffer were separated using 3-8% Tris-acetate gels (NuPAGE® Life Technologies). For single cross-links, SDS-PAG lasted 150 minutes with 150 V. For double crosslinks, SDS-PAG lasted 210 minutes with 150 V. The proteins were then transferred onto 0.2µm nitrocellulose using the Mini Trans-blot® Cell system (Bio-Rad).

To visualise the proteins, the luminant signal was developed using Immobilon™ Western Chemiluminescent HRP substrate (Millipore) and detected using G:Box Chemi-XX9 (GENESYS). The density of each band was measured using GeneTools (GENESYS). The uncropped and unprocessed scans of all the blots in the main and supplementary figures are provided in the source data file.

Crosslink efficiency was calculated using the formula-

% of crosslink=Signal for Crosslinked band/ (Signal for crosslinked band+Signal for non-crosslinked band)*100

## Calibrated ChIP-sequencing

Calibrated ChIP-sequencing was performed as previously described[27]. Forty-five mL of 15 $OD_{600}$ units of exponentially grown cells and 5 $OD_{600}$ units of reference cells *C. glabrata* were mixed with 4 mL of fixative (50 mM Tris-HCl, pH 8.0; 100 mM NaCl; 0.5 mM EGTA; 1 mM EDTA; 30% (v/v) formaldehyde) and rotated at RT for 30 min. Then 2 mL of 2.5 M glycine was added and incubated at RT for 5 min. The fixed cells were collected by centrifugation at 3500 rpm for 2 min and washed with 50 ml of ice-cold 1X PBS. The cells were then resuspended in 300 µL of ChIP lysis buffer (50 mM Hepes-KOH, pH 8.0; 140 mM NaCl; 1 mM EDTA; 1% (v/v) Triton X-100; 0.1% (w/v) sodium deoxy-cholate; 1 mM PMSF; 1 tablet/25 mL protease inhibitor cocktail (Roche)) and an equal amount of acid-washed glass beads (425-600µm Sigma) added. Cells were disrupted using a FastPrep®–24 benchtop homogeniser (M.P. Biomedicals) at 4 °C (3x 60 s at 6.5 m/s or until >90% of the cells were lysed as confirmed by microscopy).

The isolated soluble fraction was subjected to a sonication using a bioruptor (Diagenode) for 30 min in bursts of 30 s on and 30 s off at a high level in a 4 °C water bath. Under this condition,. Cell debris was removed by centrifugation at 13,200 rpm at 4 °C for 20 min and the

supernatant containing sheared chromatin with a size range of 200-1000 bp was mixed with 700 µL of ChIP lysis buffer. The samples were pre-cleared with 30 µL of protein G Dynabeads (Invitrogen) for 1 h at 4 °C. 80 µL of WCE was taken. The remaining supernatant was mixed with 5 µg of anti-PK antibody Bio-Rad) and rotated overnight at 4 °C. 50 µL of protein G Dynabeads were added and rotated at 4 °C for 2 h. The beads were washed 2x with ChIP lysis buffer, 3x with high salt ChIP lysis buffer (50 mM Hepes-KOH, pH 8.0; 500 mM NaCl; 1 mM EDTA; 1% (v/v) Triton X-100; 0.1% (w/v) sodium deoxycholate; 1 mM PMSF), 2x with ChIP wash buffer (10 mM Tris-HCl, pH 8.0; 0.25 M LiCl; 0.5% NP-40; 0.5% sodium deoxycholate; 1 mM EDTA; 1 mM PMSF) and 1x with TE pH 7.5. The immunoprecipitated chromatin was then released in 120 µL of TES buffer (50 mM Tris-HCl, pH 8.0; 10 mM EDTA; 1% SDS) by heating at 65 °C for 15 min and the collected supernatant termed the IP sample. The WCE extracts were mixed with 40 µL of TES3 buffer (50 mM Tris-HCl, pH 8.0; 10 mM EDTA; 3% SDS). The crosslink of WCE and IP samples were reverted at 65 °C overnight. RNA was digested with 2 µL RNase A (10 mg/mL; Roche) for 1 h at 37 °C and protein was degraded with 10 µL of proteinase K (18 mg/mL; Roche) for 2 h at 65 °C. DNA was purified using ChIP DNA Clean and Concentrator kit (Zymo Research).

## Preparation of sequencing libraries

Sequencing libraries were prepared using NEBNext® Fast DNA Library Prep Set for Ion Torrent™ Kit (New England Biolabs) following the manufacturer's instructions. Briefly, 10–100 ng of purified DNA was subjected to end repair and ligated with the Ion Xpress™ Barcode Adaptors. Fragments of 300 bp were then selected using E-Gel® Size-Select™ 2% Agarose gels (Life Technologies) and amplified with 6-8 PCR cycles. The DNA concentration was then determined by qPCR using Ion Torrent DNA standards (Kapa Biosystems) as a reference [12-16]. libraries with different barcodes were then pooled together to a final concentration of 350pM and loaded onto the Ion PI™ V3 Chip (Life Technologies) using the Ion Chef™ (Life Technologies). Sequencing was then completed on the Ion Torrent Proton (Life Technologies), typically producing 6–10 million reads per library with an average read length of 190 bp.

## Data analysis, alignment and production of BigWigs

Data analysis was performed on the Galaxy platform[47]. The quality of the reads was assessed using FastQC (Galaxy tool version 1.,0.0) and the low-quality bases were removed as required using 'trim sequences' (Galaxy tool version 1.0.0). Generally, this involved removing the first 10 bases and any bases after the 200th but trimming more or fewer bases may be required to ensure the removal of kmers and that the per-base sequence content is equal across the reads. Reads shorter than 50 bp were removed using Filter FASTQ (Galaxy tool version 1.0.0, minimum size: 50, maximum size: 0, minimum quality: 0, maximum quality: 0, maximum number of bases allowed outside of quality range: 0, paired end data: false) and the remaining reads aligned to the necessary gen-ome(s) using Bowtie2 (Galaxy tool version 0.2) with the default (--sen-sitive) parameters (mate paired: single-end, write unaligned reads to a separate file: true, reference genome: SacCer3 or CanGla, specify read group: false, parameter settings: full parameter list, type of alignment: end to end, preset option: sensitive, disallow gaps within *n*-positions of read: 4, trim *n*-bases from 5' of each read: 0, number of reads to be aligned: 0, strand directions: both, log mapping time: false).

To generate alignments of reads that uniquely align to the *S. cerevisiae* genome, the reads were first aligned to the *C. glabrata* (CBS138, genolevures) genome and retrieved the unaligned reads. These unaligned reads were then aligned to the *S. cerevisiae* (sacCer3, SGD) genome and the resulting BAM file was converted to BigWig using 'BAM to BigWig' (Galaxy tool version 0.1.0) and the reads are normalised as counts per million (CPM). Similarly, this process was done with the order of genomes reversed to produce alignments of reads that uniquely align to *C. glabrata*.

## Visualisation of ChIP-seq profiles

The BigWigs were visualised using the IGB browser[48]. To calibrate the ChIP profile, the track was multiplied by the samples occupancy ratio (OR) using the graph multiply function.

The Bigwigs file was used to calculate the average occupancy of the 10 kb region around all 16 centromeres through the tool "computeMatrix" and OR was used as the scale factor. A BED file named "yeast CDEIII" is used to define the centres of all the 16 CDEIII in this calculation (Supplementary Table 4).

## Protein purification of the cohesin and Scc2 complexes

The codons-optimised *SMC1/8xHis-SMC3* or *smc3-QQ/SCC1-2xStrepII/SCC3* and *GFP-ΔN132-scc2-1xStrepII* were cloned into bacmids and transfected into Sf9 cells (ThernoFisher, CAT# 11496015) using Fugene HD reagent (Promega). The generated viruses were infected into Sf9 cells, and the cells were cultured at 27 °C for 72 h in Insect-XPRESS protein-free medium with L-glutamate (Lonza). To express the proteins, 500 ml of SF-9 insect cells with a cell density of ~3 million/ml were infected with the appropriate baculovirus stock in a 1/100 dilution and harvested when lethality (assayed by the trypan blue test) reached no more than 70-80%. Cell pellets were quickly frozen in liquid nitrogen and stored at −80 °C. The thawed pellets were suspended in ~65–70 ml of HNTG lysis buffer (25 mM Hepes pH 8.0, NaCl 150 mM, TCEP-HCl 1 mM and Glycerol 10%). The suspension was immediately supplemented with 2 dissolved tablets of Roche Complete Protease (EDTA-free), 75 µg of RNAse I and 7 µl of DNAseI (Roche, of 10U/µl stock). The cells were then sonicated at 80% amplitude for 5 s/burst/35 ml of suspension using a Sonics Vibra-Cell (3 mm microtip). In total 5 bursts were given for every 35 ml half of the 70 ml suspension (the sonication was always performed in ethanolised ice). Cell debris were removed using Ti45 fixed angle rotor (45,000 rpm x 45 mins). A final concentration of 2 mM EDTA was added to the cleared extract and loaded to a 2x5ml StrepTrap HP (Fisher Scientific) column at 1 ml/min in an ÄKTA Purifier 100. Washed with HNTG containing 1 mM PMSF and 2 mM EDTA (HNTGPE) with a flow speed of 1 ml/min until ΔAU$_{280nm}$ is reduced to ~0. The protein was eluted using HNTGPE + 20 mM desthiobiotin (Fisher Scientific) at 1 ml/min. Peak fractions analysed using SDS-PAGE were pooled and were further purified in a Superose 6 Increase 10/300 (VWR) using HNTG as a running buffer (free of EDTA/PMSF). The resulting peaks were again analysed using SDS-PAGE and the concentration was determined by measuring A280 using Nanodrop. Protein was aliquoted and stocked typically in concentrations ranging from 1 to 3 mg/ml.

## ATPase assay

ATPase activity was determined as described in the protocol of the EnzChek phosphate assay kit (Invitrogen). The tetramer cohesin of a final concentration of 50 nM was dissolved with 50 mM NaCl. If dsDNA is presented, 700 nM 40 bp dsDNA was used. The reaction was initialised by adding ATP to a final concentration of 1.3 mM (final reaction volume: 150ul). The OD at 360 nm was monitored every 30 s for 90 min using a PHERAstar FS. ΔAU at 360 nm was translated to Pi release using an equation derived by a standard curve of KH$_2$PO$_4$ (EnzChek kit). Rates were calculated from the slope of the linear phase (first 10 min). At least two independent biological experiments were performed for each experiment.

## Structure modelling

The model of an Smc1-Smc3 heterodimer with coiled-coil extensions from the ATPase head domains was created by a combination of the Smc1 homodimer structure (PDB 1W1W) and the Smc3:Scc1 complex structure (PDB 4UX3) as outlined previously[28,49]. The model was developed to be consistent with the cross-linking data using the program Chimera[50] by manual adjustment of the locations of the coiled coils for each Smc protein, using the Smc3 structure as a template and correcting for the sequence of the Smc1 protein chain. The coiled-coil sections were aligned such that the positions of residues identified by the cross-linking studies were consistent with distances required to make covalent cross-links and such that allowed values for main chain rotation angles were still achieved.

## Statistics and reproducibility

All the experiments are repeated at least twice and the results of repeats are consistent. To quantify the crosslinking efficiency based on Western blot, the experiments were repeated at least three times using independent cultures. The results indicating crosslinking efficiency were plotted as the mean ± SD using Minitab Statistical Software. No data were excluded from the analyses.

## Reporting summary

Further information on research design is available in the Nature Portfolio Reporting Summary linked to this article.

## Data availability

The data that support this study are available from the corresponding author upon request. The calibrated ChIP-seq data (raw and analysed) were deposited to GEO under accession code GSE217833. All the other original/analysed data including the modelled structures were deposited to figshare: https://doi.org/10.6084/m9.figshare.22664902.v2. The Cryo-EM strucuter of Scc2/J-cohesin was obtained from EMD (EMD-12880). The Cryo-EM strucuter of Scc2/E-cohesin/DNA was obtained from PDB (6ZZ6). The crystal structure of Smc3-Scc1 was obtained from PDB (4UX3). The crystal structure of Smc1-Scc1 was obtained from PDB ((1W1W). Source data are provided with this paper.

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

## Acknowledgements

The authors are grateful to Hu and Nasmyth lab members for useful discussions, A Donaldson and A Lorenz for critical reading of the manuscript. B Hu was supported by BBSRC (BB/S002537) and the Wellcome Trust (202062/Z/16/Z).

## Author contributions

A.K. and B.H. designed and conducted experiments and wrote the manuscript. T.T. designed and conducted experiments. N.J.P. performed calibrated ChIP-seq and revised the manuscript. M.V. purified cohesin tetramer/Scc2 and conducted ATPase analysis. C.P. and P.D. conducted experiments. J.B.R conducted protein structure modelling. K.A.N. designed experiments and wrote the manuscript.

## Competing interests

The authors declare no competing interests.
