## [Peer Review File · Nature Communications]

Conformational dynamics of cohesin/Scc2 loading complex are regulated by Smc3 acetylation and ATP bindingREVIEWER COMMENTS

Reviewer #1 (Remarks to the Author):

Cohesin plays a crucial role in the proper functioning of sister chromatids during cell division and the organization of DNA during interphase. However, our understanding of how cohesin performs these functions is limited. The Hu lab's new study delves into the dynamic protein configurations involved in the loading of cohesin onto DNA in yeast. The study begins by identifying mutations that can suppress the lethal effects of acetylation-mimicry mutations on Smc3. Using a combination of genetics and site-specific chemical cross-linking, the researchers thoroughly examine the consequences of these mutations. Ultimately, they identify a previously unknown intermediate state of cohesin, called "pre-E" cohesin, which seems to be crucial for DNA loading. Acetylation of Smc3 inhibits the formation of the pre-E cohesin state. The authors propose that Smc3 KK residues (but not their acetylated forms or QQ) help stabilize the interactions between Scc2 with pre-E cohesin and E-cohesin, thereby facilitating DNA loading. Overall, this study is an excellent study that was carried out effectively. The main findings are well supported by the data. The manuscript sets the stage for determining a structure of the new "pre-E" state of cohesin and other SMC complexes. The figures are also effective but would benefit from more consistent labelling of panels and mutations. The comments or issues raised below should be addressed prior to publication.

Major comments

The authors assert that QQ does not affect DNA-stimulated ATP hydrolysis by cohesin based on in vitro experiments using a specific concentration of DNA (40 bp DNA at 700 nM). To corroborate this observation, the authors should perform similar experiments with reducing DNA concentrations to determine if QQ may actually reduce the apparent affinity for DNA.

The double cross-linking shown in Fig 4A and B are key experiments for the conclusions of the study. As control, the efficiency of head cross-linking for E-cohesin and J-cohesin should be shown in addition to the BPA cross-linking.

In Line 245 and below, the wrong unit for residue distance is used. The distance should be 16 Å, not 16 nanometers (nm). Importantly, this number is not entirely out of range for BMOE cross-linking, given the structural flexibility of the proteins. It should also be explained how the distance values were obtained in the first place (using C-alpha or C-beta distance?). Additionally, the figures include different values for the same distances (in large font size and small font size). These issues should be corrected.

There are several mentions of "data not shown" in the manuscript. These should be removed, or the data should be included.

It is possible that some of the findings are also explained by an alternative scenario in which acetylated Smc3 KK residues (but not the non-acetylated form) stabilize J-cohesin and thus reduce the prevalence of pre-E and E-cohesin. This possibility should be considered in the discussion, or the authors should explain why it is unlikely. Without access to structural models (including for J-cohesin), it is difficult to evaluate the validity of this scenario. It would thus be helpful if the authors could share their structural models (even if they are crude) with reviewers and eventually readers.

Some of the experiments have limitations that should be addressed. For example, Smc3-QQ cohesin is not bound to chromosomes (much), so the levels of E-cohesin head cross-linking may be indirectly affected by its mislocalization. In other cases, alterations in cell cycle progression may impact the results (Figures 1C and 1H). These issues should be discussed in the manuscript. The authors may want to highlight the similarities and differences between this study and research on other SMC complexes and provide some insight into the relevance of their new findings for other SMC complexes.

Minor comments:

Whenever possible, the relevant PDB code should be provided (in addition to the citation).

Denote whether experiments have been done with purified proteins or in vivo in the figure and mention in main text to avoid confusion.

Consistently label gel and blot images (anti-Scc2...) in main and supplemental figures.

Fig 4A/B: M47 or M74?

Fig 4B and S4E: A positive control for cross-linking with R830C would be helpful. At the least, the expression of the mutant protein should be confirmed by Western blotting.

Fig. 5F: denote Scc2 mutation in figure panel. Same for Smc3 in Fig. S4E.
Line 46: remove 'entrapment'
Line 72: replace 'whether' by 'how'?
Line 86: 'clamp' should be 'clamping' ?
Line 180: replace 'these three regions with 'the following three regions'
Line 195: replace 'indicating' by 'as expected for a'
Line 259: 'predicts....facts' is contradictory. Rewrite!
Line 305: 'reflects' should be 'likely reflects'
Line 318/319: confusing sentence, please rewrite
Line 324: 'the only Scc2/cohesin interface' should this be J-cohesin here?
Line 405-407: unclear statement, rewrite?
Line 427: remove 'novel' (not necessary here).
Line 451: 'Scc1 interacts WITH cohesin'
Line 479: ATP may be bound to unengaged half-sites?
Line 499: 'this mutation largely restores the DNA association of Smc3QQ but has little effect on release' clarify the statement on release. Release of otherwise wt Smc3? Non QQ?

Line 952: what is the 'oscillation model'?

Reviewer #2 (Remarks to the Author):

Hu and colleagues describe the identification of a new conformational state in the ATPase reaction cycle of the Scc2-bound cohesin complex, which they identify through a combination of yeast genetics and elegant yet complex protein-protein crosslinking experiments. The authors suggest that the newly identified state might represent an intermediate between the ATP-free ('juxtaposed') and ATP-bound ('engaged') conformations that had previously been resolved by cryo-EM. The authors provide evidence that this state plays a role in loading cohesin onto chromosomes, which is counteracted by acetylation of Smc3.

The molecular details of how cohesin complexes load onto chromosomes and how this reaction is controlled through regulatory components (such as Pds5, Wapl, Eco1) has remained largely unknown. I highly appreciate the authors' efforts to gain mechanistic insights through the identification of meaningful suppressor mutants and by capturing dynamic conformations in the cohesin complex that might not be accessible to conventional structural biology approaches, for example due to their transient nature. The authors have screened dozens of different sites in different cohesin subunits for their ability to generate crosslinks with other subunits and have performed several genetic suppressor screens. In this respect, the work described in this manuscript is a true tour-de-force.

Nevertheless, I'm not convinced from the data presented in this manuscript that the state the authors claim to have identified is different from the previously identified engaged state (major comments 1-7), nor that this state, should it exist, were involved in DNA loading (major comment 8). Given the complexity of many of the yeast experiments and the difficulty in interpreting the results of these experiments in a straight-forward way (major comments 9-10), it is obvious that the manuscript addresses a specialized audience, while some of the key questions (e.g., what is the functional role of Smc3 acetylation?) remain unanswered. I cannot envision a revised form of this manuscript that would be suitable for publication in Nature Communications. The comments below are intended to help the authors to improve their manuscript for publication elsewhere.

Major comments:

The authors provide several arguments why they think that the conformational state they largely deduce from crosslinking of a single residue pair (Smc3 E67 - Scc2 T1175) is different from the ATP-engaged state previously observed in cryo-EM structures (Shi et al., 2020; Higashi et al., 2020; Collier et al., 2020). Many of the other arguments are largely based on negative results, i.e., the absence of crosslinks.

1. The authors claim that the „distances between Smc3 Q67 and Scc2 T1175 are more than 16nm“ in the engaged structure, which is incompatible with BMOE crosslinking (line 245). This is incorrect. The Calpha-Calpha distance between these two residues is 1.49 nm in the structure of the ATP-engaged *S. cerevisiae* cohesin complex (pdb 6zz6; Collier et al., 2020) and is hence within range of crosslinking, particularly when taking into account that Smc3 Q67 is located in a flexible loop region. Rotation of the side chain or a minor repositioning of the loop would suffice to allow a BMOE crosslink to occur. Such motions are compatible with the reported electron density map (EMD-11585).

2. The authors argue that the Smc1-Smc3 head domains cannot be associated with each other when Smc3 Q67 crosslinks to Scc1 because they would have expected that three Smc1 residues within the vicinity of Smc3 Q67 crosslinked to Scc2 if the head domains were engaged. However, there are two Smc3 residues (Smc3 R58 and L111) within the same distance of Smc3 Q67 (11.5-12.5 Å, Calpha-Calpha, pdb 6zz6) as the three Smc1 residues (9.6-14.1 Å, Calpha-Calpha). These Smc3 residues also don't crosslink to Scc2. There might hence be other reasons why certain residues cannot be crosslinked with high efficiency.

3. Smc3 M74 crosslinks with equally low efficiency to Scc2 E821 as it does to N886 (and another five residues within the same patch). The authors argue that the lack of Smc3 M74 crosslinking to Scc2 R830 is evidence for two discrete conformations (lines 256-258). Yet, again the authors must consider that a lack of BMOE crosslinking can be due to a variety of reasons, including the rigid positioning of R830 in the middle of an alpha-helix, as opposed to the more flexible positions of Scc2 residues E821 and N886. A preferred crosslinking to the nearby C872 residue could also explain a lack of observed crosslinks to Smc3 Q67C. Flexibility of the Smc3 contact loop cannot be excluded based on this data and all observed crosslinks are compatible with the previously reported cryo-EM structure of engaged cohesin.

4. In the region of Scc2 that crosslinks to Smc3 Q67, Scc2 residues S1171 and T1175 are the closest to Smc3 Q67 in the cryo-EM structure of the ATP-engaged state (pdb 6zz6). Similarly, Scc2 T1281 is very close to Smc1 E1102 in this structure. The authors' crosslinking results are therefore consistent with the engaged state, which also can explain why both interfaces simultaneously crosslink to Scc2.

5. The absence of Smc1-Smc3-Scc2 crosslinks after BMOE crosslinking of the Smc1-Smc3 heads followed by Smc3 Q67 crosslinking (Fig. 4D) must be interpreted carefully. First, it is unlikely that the BMOE crosslink will prevent ATP hydrolysis and ADP release of the engaged heads, and the resulting cohesin complex – though crosslinked – cannot be equated to the properly engaged conformation in which ATP is sandwiched between the ATPase domains of Smc3 and Smc1. Second, the fraction of Smc1-Smc3 BMOE crosslinks detected in these experiments is minimal (Fig. 4D, right gel, lane 3). It is therefore not surprising that no Smc1-Smc3-Scc2 crosslinked species can be detected. In contrast, it appears that the efficiency of Smc3-Scc2 crosslinking is not reduced in the presence of BMOE (compare lanes 2 and 4). The reason why a triple crosslink can be detected for the Smc1 E1102 BPA construct (Fig. 4D, left gel, lane 4) is probably that more sample had been loaded in this lane (compare the non-crosslinked Scc2 bands).

6. The finding that the Smc3 QQ mutation reduces crosslinking of the Smc3 E67 residue (Figure 5) to the same extent as it reduces crosslinking of two residue pairs specifically designed for probing the engaged state (Smc3 S72 - Scc2 E819 and Smc3 K1004 - Scc2 D369) furthermore supports the conclusion that the E67 residue in fact detects the engaged and not a previously undescribed state.

7. The authors claim that locking the Smc1-Smc3 heads in the engaged state by Walker B mutations increases the crosslinking efficiency of Smc3 E67 to a lesser degree than it does for two combinations designed to probe the engaged state (Figure 7A). In the absence of any quantitation of the data presented in this figure, this conclusion is hardly justified.

In general, the authors should provide quantitative data for multiple repeats of each crosslinking experiment instead of just presenting a single western blot. When the authors present quantitative

data, it is not clear how they corrected for different blotting efficiencies of non-crosslinked and crosslinked species. Comparing the decrease in the non-crosslinked species would probably be the most accurate way to compare different crosslinking reactions. Such comparisons are essential, for example for Figure 2C, where the non-crosslinked band of Smc3 K38I appears significantly reduced when compared to the non-crosslinked band of wild-type Smc3.

8. The authors suggest that the interaction between Scc2 and the Smc3 coiled coil is weaker in the pre-engaged state than in the engaged state and that this interface therefore acts as an entry gate for DNA (lines 481-483). This conclusion is presumably based on the different crosslinking efficiencies of the Smc3 K1004 and Scc2 D369 residues in the Walker B mutant Smc3 (Figure 7A). Can this conclusion really be drawn from the available data? Even if distinct pre-engaged and engaged conformations existed, it is likely that the former is much more transient than the latter. Differences in crosslinking efficiencies therefore don't necessarily reflect the stability of protein-protein interactions.

9. Several conclusions rely on the interpretation of yeast strains with four or more mutations, for example smc3 K112Q K113Q R1008I scc2 E822K L937F (Figure 6F-I). Drawing specific mechanistic conclusions from such mutant combinations needs to be done with particular care. For example, the authors first suggest that the Smc3 R1008I mutation suppresses the Smc3 QQ phenotype by stabilizing the Scc2-Smc3 coiled coil interaction (lines 350-351) but later suggest that this mutation instead destabilizes juxtaposed coiled coils (lines 554-556). It remains also unclear how the W483R mutation located close to the hinge domain of Smc3 restores the (pre-)engaged state in the Smc3 QQ mutant. The authors speculate that this mutation might, like R1008I, destabilize the juxtaposed coiled-coil conformation. This suggestion could be directly tested with the authors' crosslinking assay.

10. In their interpretation of the experiments with acetylation-mimicking Smc3 QQ mutants, the authors ignore a potential function of the Pds5 protein, which is known to bind to these residues (Petela et al., 2021) and enacts its downstream effects on cohesion establishing (Hartman et al., 2000) and Wapl-mediated unloading (Shintomi & Hirano, 2009). Pds5 and Scc2 bind cohesin in a mutually exclusive manner, and a lack of Smc3 Q67 crosslinking to Scc2 is an expected result of Scc2 replacement by Pds5. Indeed, the reduced detection of Smc1-Scc2 binding in the immunoprecipitated complexes (Figure 5J) are not explained by the pre-engaged, engaged, and juxtaposed conformations. As Pds5 relies on a folded elbow to bridge the acetylated patch on the Smc3 ATPase head and the hinge domain (Petela et al., 2021), mutations that are expected to destabilize elbow folding, such as Smc3(W483R) could also reduce Pds5 binding. The authors should probe for Pds5 in their immunoprecipitation experiments (Figure 5J) and repeat the BMOE crosslinking experiments of Smc3 QQ mutants (Figures 5A, F, H and 6A, C) with purified cohesin in vitro to exclude secondary effects of interference of Pds5 in vivo. In vitro experiments also enable the use of non-hydrolysable ATP analogues as an alternative to sequential BMOE and BPA crosslinking (Figure 4D).

Considering that Smc3 Q67 crosslinks to Scc2 are compatible with the structure reported for DNA-bound engaged cohesin (Collier et al., 2020), a simpler interpretation of the data in Figure 4 is that DNA-binding of the mutant is much reduced (Figure 1C) and hence fewer complexes can be found in the DNA-bound engaged conformation, either probed for by Smc3 Q67 (Figure 4B), S72 (Figure 4G), or K1004 (Figure 4I).

Minor comments:

11. The abstract should mention that the study was performed in yeast.

10. Line 150: To compare ATP hydrolysis rates in the presence of Scc2 and DNA, the authors should determine hydrolysis rates (ATP molecules hydrolysed per second per cohesin molecule) from the linear part of the curves shown in Figures 2A and B and determine average rates from independent repeats of the experiment. The same comment applies to Figures 6 E, H, and I.

12. Line 186: Q66 should be H66.

13. Lines 515-516: "reversing the positive charge of Scc2 E822 leads to a gain of-function allele of Scc2 (scc2-E822K)". The authors probably wanted to write that the mutation reversed the negative charge of glutamate.

13. All figures with Western blots: it would be extremely helpful if the authors indicated size markers on each blot to allow the reader to compare the assignment of crosslinked species. Some of the gels have been extensively cropped (e.g., Figure 2E).

14. Figure 1C: Blue and green curves almost indistinguishable.

15. Figure 1D: W483R is mislabelled as W384R.

16. Figure 1E: 483R should be W483R.

17. Figure 1G: A smc3QQ delta-eco1 control strain should be included in the experiment (which should be viable).

18. Figure 4B: M74C is mislabelled as M47C.

19. Language editing will be required throughout the manuscript (for example, lines 478-479: "the heads in the Scc2/pre-E-cohesin complex are disengaged, which is unable to bind ATP."; lines 522-524: "so smc3QQ is still able to form a complex with Scc2 interaction, which is expected in the Scc2/J-cohesin configuration"; lines 933-935: "After immunoprecipitated against Scc1-PK and separated on gradient gel, Scc2...").

20. Even if the manuscript addresses a specialized audience, the authors nevertheless need to outline their experimental strategies. For example, the crosslinking experiment shown in Figure 1H is impossible to interpret without some additional explanation in the figure legend. In general, additional information could be provided in the figure legends.

21. The authors need to carefully check their methods section. For example, were yeast cells really incubated shaking at 2000 rpm (line 585)?

22. Both supplementary tables are labelled as S1.

Reviewer #3 (Remarks to the Author):

The manuscript by Kaushik et al., extends our knowledge of the molecular mechanism by which the chromosome-organising complex, cohesin functions. The authors use elegant and sophisticated cross-linking experiments in budding yeast together with structural predictions to provide evidence for the existence of cohesin-Scc2 configurations in vivo. These configurations include two configurations previously identified through in vitro structural studies (Scc2 in complex with E-cohesin and J-cohesin), providing crucial evidence that these conformations exist also in vivo. The authors additionally identify a new configuration, which they name "pre-E-cohesin", which is likely a key intermediate in the structural transition between E- and J-cohesin. The authors go on to show, using an Smc3 acetylation mimicking mutation (smc3-QQ), that Smc3 acetylation prevents Scc2 binding to E-cohesin and pre-E cohesin. Using a genetic screening approach, the authors additionally identify mutations in SMC3 that overcome the lethality of the Smc3-QQ. They show that these smc3-QQ-suppressing mutations enhance Scc2 binding to E-cohesin and pre-E cohesin, allowing them to link the structural transitions of cohesin to cellular function (cohesin levels on chromosomes). This is a high quality, extremely thorough and insightful study with rigorously performed experiments and important conclusions. The findings will be of great interest to the chromosome biology and molecular motor fields. The experiments and data are beautiful. However, the manuscript needs significant re-writing to make the findings accessible and to ensure that the study has the impact it deserves.

Major concerns

1. The manuscript is very dense and challenging to decipher. Please re-write with clear explanations for the non-specialist reader. There are many places where more explanation is needed or where sentences do not make sense. The most prominent are listed in "minor concerns".
2. Figure legends are sparse and lack key information. Please provide more details of the experiments (which antibody?, are the blots from whole cell extracts etc.) to allow the reader to fully understand the experiments performed.
3. On many figures the antibody used for western blotting is not labelled which makes the figure difficult to interpret. Please ensure that it is clear how and what is being detected on the blots.
4. The rationale for the screen in which *smc3E199A* and *smc3R10081* were identified is not clear. These mutations were identified as suppressors of the *wpl1Δ* and *scc2-45* synthetic lethality. First, it is not clear why the absence of the Wpl1 cohesin-destabilising protein would be synthetic lethal with *scc2-45* where less cohesin should be loaded. Second, it is not clear why such a suppressive mutation would be expected to rescue *smc3-QQ*. Is the rationale that *wpl1Δ scc2-45* is not viable because *scc2-45* relies on cohesin recycling by Wpl1 to have sufficient cohesin for cohesion? And that *smc3-QQ* would similarly reduce the availability of cohesin for loading? Please provide an explanation for the logic behind this screen and its application to *smc3-QQ*.
5. The experiments presented in Figure 5 make the assumption that the interaction between Scc2-Smc3 CC is a marker of engaged heads (E-cohesin). However, the evidence that shows that this is the case is only presented in Figure 7. A more logical presentation of the data would therefore be to present the data in Figure 7 before current Figure 5.
6. The authors use their elegant cross-linking approach to probe specific interactions of Scc2 with Smc3QQ and conclude by a process of elimination that J-cohesin can still bind. However, they cannot rule out that Smc3-QQ could bind distinct cohesin conformations that are yet to be discovered. Though not essential, it would strengthen the manuscript to determine whether it Scc2 can really associate with Smc3-QQ J-cohesin. Could the authors lock Smc3-QQ cohesin in the J- or E conformation (as in Figure 4) to address this point?
7. Throughout the manuscript, including the introduction and discussion, the emphasis is on cohesin loading. However, Scc2 is also driving loop extrusion and this should also be discussed in the context of what they found. Only the rescue of viability can be specifically connected to loading.
8. The ATPase assays in Figure 6H and I: are they with DNA or not? This information needs to be included in the figure and legend. If there is DNA why is the result different from the wild type data in Figure 2A. If there is DNA, why does the wild type show such a strong response in Figure 6H but not 2A? If there is DNA, why doesn't the QQ mutant show a stronger response in Figure 6I.

Minor comments

1. Figure S1A. It would be useful to include this in the main figures for ease of understanding.
2. Page 2, line 46. This sentence is difficult to understanding and too simplistic. The authors should discuss the findings from Bauer 2021 in their manuscript, in particular the knowledge that cohesin has more than 2 configurations and that it undergoes cyclical structural changes.
3. Page 3, line 82. "corroborating" does not fit here. Perhaps the authors could simply say that they identified a new configuration in addition to the two published ones? However, see point 2 above.
4. line 85. Should read "the DNA clamp". However this is the first mention and the DNA clamp has not been described. Some introduction/explanation is needed. As from the points above, a more

general and complete description of cohesin structural transitions/models is required.

5. Line 104. "evolved" suggests an evolution experiment, while the authors actually carried out a random mutagenesis experiment. The authors could simply state that they aimed to identify intragenic suppressors by random mutagenesis.

6. Line 105. Where were the other mutations?

7. Line 131. This is the first mention of BMOE. Please explain the approach.

8. Line 144 and throughout, please correct nomenclature. Genes in italics (capitals for wild type and lowercase for mutations), proteins in titlecase non-italic. There are many instances in the figures and text where this is incorrect.

9. Line 182 "KKD strand". It is not immediately obvious what this means or that this is the residues that are acetylated by Eco1. More explanation is required. It is also not necessary to abbreviate unless it will be used again later.

10. Line 184. Could the figure be improved to make the potential interaction surfaces easier to understand?

11. Figure 3B. The inability to detect the interaction does not mean that they do not interact - perhaps the substitutions are not appropriately positioned. The structure does show the interaction so this interaction seems likely. The interpretation of this experiment should be adjusted to account for this.,

12. Figure S3A and throughout. Figure and figure legends need to state which antibodies used for western blot. More details of the assay are required. Is this whole cell extract or immunoprecipitated protein?

13. Line 209. Summary sentence is not very informative. It would be useful for the reader to state what the two interfaces are.

14. Figure S3D. This is an elegant experiment but the figure is laid out in a very confusing way. Flag is coloured in green but on the blot, the green dots refer to the lack of Flag. Also the westerns are flipped so that the HA blot is aligned with schematic which includes Flag, it would be better to reverse them. Also label the blot with antibody used - this is critical to understand the experiment.

15. Figure 3H. It would be helpful to indicate the residues used to generate cysteine pairs in a close up of this model.

16. Line 249. Not really a neighbour residue. Could the authors show this is Figure S4A/B?

17. Line 259. "predict a fact" is awkward. " "Makes two predictions" would be better.

18. Line 260. The phrasing here is difficult to understand. The authors should say that the model predicts that the CC are separate.

19. Figure 4C. Please label on the plot that this is a three protein interaction for ease of understanding. Smc3-Scc2-Smc1. It is not correct to call it J-cohesin as this is not what is on the blot after denaturing.

20. Line 295. Please indicate in the figure that this is Scc2-T1175C.

21. Figure 5C. It is difficult to discern Scc2 and Smc3 in this figure as the colours are very similar, although it is appreciated to use the same colour scheme throughout the manuscript. Consider changing the colours. The schematics showing positions of cross-links are extremely useful.

22. Line 334. "This conclusion was supported.. (data not shown)." This sentence does not provide

any additional clarity/confirmation without better explanation. Either delete the sentence or show the data.

23. Line 336. The conclusion that Scc2 interacts with this interface in the J configuration is too strong because there may be other configurations that we don't know about. It also cannot be ruled out that Scc2 interactions with 1102 in the E or pre-E configurations but less efficiently. Although not required, this could be tested by locking cohesin in the J configuration and then testing binding.

24. Figure 6C. DO the authors have a stronger exposure of this blot? It is difficult to see the bands referred to.

25. Figure 6E. The authors should show wild type and smc3-QQ controls for comparison.

26. Line 362. The summary is confusing. Do the authors mean to provide an explanation for why loading is rescued, even though Scc2 only mildly stimulates ATPase activity? Is this explanation the idea that because Scc2 binds more efficiently to Smc3QQ this brings the complex to the DNA, allowing further stimulation of ATPase?

27. Please call out figures in order in the text.

28. Figure 6 legend, letters are mixed up.

29. In Figure 7, are the authors using the EQ mutant for both Smc1 and Smc3? If so, this should be indicated in the figure.

30. The discussion should be shortened as the main points are lost. The authors are encouraged to provide concise discussion focused on three points: 1. The pre-E state they identified. 2. Confirmation of in vivo existence of configurations predicted from structural studies. 3, How acetylation affects Scc2 interaction with cohesin.

31. In the methods "collect 30 OD of cells" does not make sense. What volume?

This document contains reviewers' comments in bold red, authors' responses in black and mentions of additional data in green.

Reviewer #1 (Remarks to the Author):

Cohesin plays a crucial role in the proper functioning of sister chromatids during cell division and the organization of DNA during interphase. However, our understanding of how cohesin performs these functions is limited. The Hu lab's new study delves into the dynamic protein configurations involved in the loading of cohesin onto DNA in yeast. The study begins by identifying mutations that can suppress the lethal effects of acetylation-mimicry mutations on Smc3. Using a combination of genetics and site-specific chemical cross-linking, the researchers thoroughly examine the consequences of these mutations. Ultimately, they identify a previously unknown intermediate state of cohesin, called "pre-E" cohesin, which seems to be crucial for DNA loading. Acetylation of Smc3 inhibits the formation of the pre-E cohesin state. The authors propose that Smc3 KK residues (but not their acetylated forms or QQ) help stabilize the interactions between Scc2 with pre-E cohesin and E-cohesin, thereby facilitating DNA loading.

Overall, this study is an excellent study that was carried out effectively. The main findings are well supported by the data. The manuscript sets the stage for determining a structure of the new "pre-E" state of cohesin and other SMC complexes. The figures are also effective but would benefit from more consistent labelling of panels and mutations. The comments or issues raised below should be addressed prior to publication.

Major comments

The authors assert that QQ does not affect DNA-stimulated ATP hydrolysis by cohesin based on in vitro experiments using a specific concentration of DNA (40 bp DNA at 700 nM). To corroborate this observation, the authors should perform similar experiments with reducing DNA concentrations to determine if QQ may actually reduce the apparent affinity for DNA.

We agree with the reviewer. Although smc3QQ's ATPase activity is largely restored by the addition of DNA, it doesn't reach the level of WT. The comparison is shown in Fig S2A. It is possible that QQ might slightly affect DNA/cohesin association. However, as we claimed in the manuscript, this is not the major reason that causes the loading defect of smc3QQ.

The double cross-linking shown in Fig 4A and B are key experiments for the conclusions of the

study. As control, the efficiency of head cross-linking for E-cohesin and J-cohesin should be shown in addition to the BPA cross-linking.

We have included the proposed data in Fig S4H and S4I.

In Line 245 and below, the wrong unit for residue distance is used. The distance should be 16 Å, not 16 nanometers (nm). Importantly, this number is not entirely out of range for BMOE cross-linking, given the structural flexibility of the proteins. It should also be explained how the distance values were obtained in the first place (using C-alpha or C-beta distance?). Additionally, the figures include different values for the same distances (in large font size and small font size). These issues should be corrected.

We thank the reviewer for pointing this out. We have resolved this issue, and the distances shown are between C-betas as it justifies the importance of orientation and distances between -SH groups. (we have now added this information to the respective sections). According to Green et al., the average distance for BMOE to cover is 6-10.52 Å. E1102-T1281 is under this distance, but Q67 is not (Green et al., 2001).

There are several mentions of "data not shown" in the manuscript. These should be removed, or the data should be included.

Thank you for pointing this out. We have removed all the "data not shown".

It is possible that some of the findings are also explained by an alternative scenario in which acetylated Smc3 KK residues (but not the non-acetylated form) stabilize J-cohesin and thus reduce the prevalence of pre-E and E-cohesin. This possibility should be considered in the discussion, or the authors should explain why it is unlikely. Without access to structural models (including for J-cohesin), it is difficult to evaluate the validity of this scenario. It would thus be helpful if the authors could share their structural models (even if they are crude) with reviewers and eventually readers.

Thanks to the reviewer for the suggestion regarding an alternative situation. However, we think this scenario is unlikely as when we checked the effect of the QQ mutant on the head engagement (fig 2C, 2D, S2), QQ mutant does increase the crosslinking efficiency of E-head. Therefore, we do not think smc3QQ stabilizes J-cohesin.

We have deposited all the structures used in this study, including the structural model for pre-E cohesin, in the figshare database.

Some of the experiments have limitations that should be addressed. For example, Smc3-QQ cohesin is not bound to chromosomes (much), so the levels of E-cohesin head cross-linking may be indirectly affected by its mislocalization. In other cases, alterations in cell cycle progression may impact the results (Figures 1C and 1H). These issues should be discussed in the manuscript.

Thanks to the reviewer for the suggestion regarding an alternative explanation. However, we think this scenario is unlikely as when we checked the effect of QQ mutants on head engagement (fig 2C, 2D, S2), QQ mutants did not inhibit head engagement. Also, to avoid the effect of the cell cycle, the *in vivo* BMOE crosslinks were performed in the cells arrested in later G1, as pointed out in the manuscript.

The authors may want to highlight the similarities and differences between this study and research on other SMC complexes and provide some insight into the relevance of their new findings for other SMC complexes.

We thank the reviewer for the suggestion and have included these points in the discussion.

Minor comments:

Whenever possible, the relevant PDB code should be provided (in addition to the citation).

Thank you for pointing this out, we have added the relevant PDB codes.

Denote whether experiments have been done with purified proteins or *in vivo* in the figure and mention in main text to avoid confusion.

All the BPA and BMOE crosslink experiments are done *in vivo*, and ATPase activity assays are carried out *in vitro*. We have added this information to the main text.

Consistently label gel and blot images (anti-Scc2...) in main and supplemental figures.

We thank the reviewer for the suggestion, and we have reformatted all figures.

Fig 4A/B: M47 or M74?

Thank this reviewer for pointing out the typo and it has been corrected. It is M74.

Fig 4B and S4E: A positive control for cross-linking with R830C would be helpful. At the least, the expression of the mutant protein should be confirmed by Western blotting.

We have added the blots for Scc2-FLAG and PGK1 in the whole cell extract, showing equal expression of Scc2 in all the strains (fig S4E).

Fig. 5F: denote Scc2 mutation in figure panel. Same for Smc3 in Fig. S4E.

We thank the reviewer for the suggestion. We have mentioned all Smc3, Smc1 and Scc2 mutations in the figure panels, including Fig S4E.

Line 46: remove 'entrapment'

We thank the reviewer for the suggestion, and we rephrased the sentence.

Line 72: replace 'whether' by 'how'?

We thank the reviewer for the suggestion, and we rephrased the sentence.

Line 86: 'clamp' should be 'clamping' ?

We thank the reviewer for pointing this out. We have corrected it.

Line 180: replace 'these three regions with 'the following three regions'

We thank the reviewer for the suggestion, and we rephrased the sentence.

Line 195: replace 'indicating' by 'as expected for a'

We thank the reviewer for the suggestion, and we rephrased the sentence.

Line 259: 'predicts....facts' is contradictory. Rewrite!

We thank the reviewer for the suggestion, and we rephrased the sentence.

Line 305: 'reflects' should be 'likely reflects'

We thank the reviewer for the suggestion. We have revised this paragraph and removed this statement.

Line 318/319: confusing sentence, please rewrite

Thank you for pointing this out. We have rewritten it.

Line 324: 'the only Scc2/cohesin interface' should this be J-cohesin here?

Thank you for pointing this out. We have rewritten it.

Line 405-407: unclear statement, rewrite?

Thank you for pointing this out. We have rewritten it.

Line 427: remove 'novel' (not necessary here).

Thank you for pointing this out. We revised the subheading.

Line 451: 'Scc1 interacts WITH cohesin'

Thank you for pointing this out. We have rewritten it.

Line 479: ATP may be bound to unengaged half-sites?

As the ABC-transporter-like ATPase domain, the Smc proteins require proper head engagement to form ATP binding pocket and hydrolyse ATP. However, as pointed out by this reviewer, it is unclear whether unengaged heads are still able to bind ATP. Therefore, we revised this statement to “unable to hydrolyse ATP”.

Line 499: ‘this mutation largely restores the DNA association of Smc3QQ but has little effect on release’ clarify the statement on release. Release of otherwise wt Smc3? Non QQ?

Thank you for pointing this out. What we meant was suppressor mutants can recover the DNA association of Smc3QQ but cannot recover the releasing defect of Smc3QQ.

Line 952: what is the ‘oscillation model’?

We have changed the sentence for a clear understanding.

Reviewer #2 (Remarks to the Author):

Hu and colleagues describe the identification of a new conformational state in the ATPase reaction cycle of the Scc2-bound cohesin complex, which they identify through a combination of yeast genetics and elegant yet complex protein-protein crosslinking experiments. The authors suggest that the newly identified state might represent an intermediate between the ATP-free (‘juxtaposed’) and ATP-bound (‘engaged’) conformations that had previously been resolved by cryo-EM. The authors provide evidence that this state plays a role in loading cohesin onto chromosomes, which is counteracted by acetylation of Smc3.

The molecular details of how cohesin complexes load onto chromosomes and how this reaction is controlled through regulatory components (such as Pds5, Wapl, Eco1) has remained largely unknown. I highly appreciate the authors’ efforts to gain mechanistic insights through the identification of meaningful suppressor mutants and by capturing dynamic conformations in the cohesin complex that might not be accessible to conventional structural biology approaches, for example due to their transient nature. The authors have screened dozens of different sites in different cohesin subunits for their ability to generate crosslinks with other subunits and have performed several genetic suppressor screens. In this respect, the work described in this manuscript

is a true tour-de-force.

Nevertheless, I'm not convinced from the data presented in this manuscript that the state the authors claim to have identified is different from the previously identified engaged state (major comments 1-7), nor that this state, should it exist, were involved in DNA loading (major comment 8). Given the complexity of many of the yeast experiments and the difficulty in interpreting the results of these experiments in a straight-forward way (major comments 9-10), it is obvious that the manuscript addresses a specialized audience, while some of the key questions (e.g., what is the functional role of Smc3 acetylation?) remain unanswered. I cannot envision a revised form of this manuscript that would be suitable for publication in Nature Communications. The comments below are intended to help the authors to improve their manuscript for publication elsewhere.

Major comments:

The authors provide several arguments why they think that the conformational state they largely deduce from crosslinking of a single residue pair (Smc3 E67 - Scc2 T1175) is different from the ATP-engaged state previously observed in cryo-EM structures (Shi et al., 2020; Higashi et al., 2020; Collier et al., 2020). Many of the other arguments are largely based on negative results, i.e., the absence of crosslinks.

1. The authors claim that the „distances between Smc3 Q67 and Scc2 T1175 are more than 16nm” in the engaged structure, which is incompatible with BMOE crosslinking (line 245). This is incorrect. The Calpha-Calpha distance between these two residues is 1.49 nm in the structure of the ATP-engaged *S. cerevisiae* cohesin complex (pdb 6zz6; Collier et al., 2020) and is hence within range of crosslinking, particularly when taking into account that Smc3 Q67 is located in a flexible loop region. Rotation of the side chain or a minor repositioning of the loop would suffice to allow a BMOE crosslink to occur. Such motions are compatible with the reported electron density map (EMD-11585).

Thank you for pointing this out. We have used C_{β} distances in our study as crosslinks occur between R-groups (SH-group for cysteines) and C_{β} more precisely reflect distances between R-groups. However, According to Green et al., the average distance for BMOE to cover is 6-10.52 Å. E1102-T1281 is under this distance, but Q67 (16.5 Å) is not (Green et al., 2001).

While we do not contradict the fact that Smc3 Q67 is in a flexible loop region, we argue that the conformation we saw in our study is not the head engagement conformation because of the following reasons:

1. We did a screen using BPA crosslink and TEV cleavage to identify the crosslink between Q67 and Scc2, and our TEV cleavage experiment showed that Q67 crosslinks to a region between Scc2 1110-1176. However, in E-state, Q67 is closer to other residues like Scc2 Y1279(6.9Å), K1278(8Å), P1246 (12.3Å) , K1240 (13.7Å). This is consistent with our claim 'Smc3 Q67/Scc2 T1175 interaction represents a configuration different from Scc2/E-cohesin'. We included this in our discussion now.
2. In this revision, we present another critical piece of evidence of the Smc3 Q67-Scc2 T1175 interface being in a different conformation than E-conformation. In figure 7B and 7C, we have presented double crosslink profiles of (a)Smc3 Q67C-Scc2 T1175C and Smc3 K1004C-Scc2 D369C and (b) Smc3 S72C-Scc2 E819C and Smc3 K1004C-Scc2 D369C. The stoichiometric ratio of Smc3 Q67C-Scc2 T1175C to Smc3 K1004C-Scc2 D369C is much higher than Smc3 S72C-Scc2 E819C to Smc3 K1004C-Scc2 D369C, reflecting Smc3 Q67C-Scc2 T1175C and Smc3 K1004C-Scc2 D369C interface being in different conformation. Also, the efficiency for the double crosslink band formed in Smc3 Q67C-Scc2 T1175C and Smc3 K1004C-Scc2 D369C is lower than the double crosslink band formed in Smc3 S72C-Scc2 E819C and Smc3 K1004C-Scc2 D369C, which supports Smc3 Q67-Scc2 T1175 and Smc3 K1004-Scc2 D369 interfaces being in two different conformations.

2.The authors argue that the Smc1-Smc3 head domains cannot be associated with each other when Smc3 Q67 crosslinks to Scc1 because they would have expected that three Smc1 residues within the vicinity of Smc3 Q67 crosslinked to Scc2 if the head domains were engaged. However, there are two Smc3 residues (Smc3 R58 and L111) within the same distance of Smc3 Q67 (11.5-12.5 Å, Calpha-Calpha, pdb 6zz6) as the three Smc1 residues (9.6-14.1 Å, Calpha-Calpha). These Smc3 residues also don't crosslink to Scc2. There might hence be other reasons why certain residues cannot be crosslinked with high efficiency.

Contradicting the reviewer's comment, we would like to point out fig no 3B, where we showed that Smc3 R58 could crosslink to Scc2. But as the reviewer said, Smc3 L111 cannot crosslink to Scc2. We could not detect any Smc3 L111BPA-Scc2 crosslink probably because this interaction would

be highly sensitive to ATP-mediated head engagement, which is transient in WT cohesin. We have included this point in the manuscript now.

3. Smc3 M74 crosslinks with equally low efficiency to Scc2 E821 as it does to N886 (and another five residues within the same patch). The authors argue that the lack of Smc3 M74 crosslinking to Scc2 R830 is evidence for two discrete conformations (lines 256-258). Yet, again the authors must consider that a lack of BMOE crosslinking can be due to various reasons, including the rigid positioning of R830 in the middle of an alpha-helix, as opposed to the more flexible positions of Scc2 residues E821 and N886. A preferred crosslinking to the nearby C872 residue could also explain a lack of observed crosslinks to Smc3 Q67C. Flexibility of the Smc3 contact loop cannot be excluded based on this data and all observed crosslinks are compatible with the previously reported cryo-EM structure of engaged cohesin.

Smc3 M74C crosslinks to Scc2 E821C with an equal efficiency as Smc3 Q67C-Scc2 T1175C and Smc3 S72C-Scc2 E819C. We provide evidence showing similar crosslink efficiency of Smc3 Q67C/S72c/M74C with Scc2 here.

We beg to differ with the reviewer about R830 not being able to be available for crosslink. Scc2 R830 is not hiding, as the side chain of this residue is pointing towards the cavity and hence exposed towards Smc3 M74.

The distance between Smc3 Q67C-Scc2 C872 is 23.6Å, beyond BMOE crosslink (PDB 6zz6, C_β-C_β). Hence, we cannot expect a crosslink between these two residues, which is also supported by our BMOE crosslink, where we had Smc3 Q67C alone (fig 3E, third lane). This observation is consistent with our hypothesis and we would like to emphasise taking the paper as a whole to support our proposed model.

4. In the region of Scc2 that crosslinks to Smc3 Q67, Scc2 residues S1171 and T1175 are the closest to Smc3 Q67 in the cryo-EM structure of the ATP-engaged state (pdb 6zz6). Similarly, Scc2 T1281 is very close to Smc1 E1102 in this structure. The authors' crosslinking results are therefore consistent with the engaged state, which also can explain why both interfaces simultaneously crosslink to Scc2.

We beg to differ with the reviewer. Scc2 residues S1171 and T1175 are not the closest to Smc3 Q67 in the cryo-EM structure of the ATP-engaged state (pdb 6zz6, C_β-C_β). The closest residues are Scc2 Y1279(<5Å), K1278(<7Å), P1246, K1240 (<10Å). These are a few examples. The distance between Smc3 Q67 with Scc2 S1171 and T1175 are 21.9Å and 16.5Å, respectively.

We agree with the reviewer that Smc1 E1102C- Scc2 T1281C and Smc3 Q67C-Scc2 T1175C are compatible. But we disagree on this conformation being PDB 6zz6 as our newly generated data for Smc3 Q67C K1004C-Scc2 T1175C D369C crosslink in the presence of Walker B mutants shows that Smc3 Q67C-Scc2 T1175C crosslink is mutually exclusive to head engagement and does not support the formation of the small compartment for the DNA insertion as seen in Smc3 S72C K1004C-Scc2 E819C D369C. Hence, once again, we emphasise on the fact on the presence of a pre-E conformation, where Smc1 E1102C- Scc2 T1281C and Smc3 Q67C-Scc2 T1175C simultaneously crosslink.

5. The absence of Smc1-Smc3-Scc2 crosslinks after BMOE crosslinking of the Smc1-Smc3 heads followed by Smc3 Q67 crosslinking (Fig. 4D) must be interpreted carefully. First, it is unlikely that the BMOE crosslink will prevent ATP hydrolysis and ADP release of the engaged heads, and the resulting cohesin complex – though crosslinked – cannot be equated to the properly engaged conformation in which ATP is sandwiched between the ATPase domains of Smc3 and Smc1.

Second, the fraction of Smc1-Smc3 BMOE crosslinks detected in these experiments is minimal (Fig. 4D, right gel, lane 3). It is therefore not surprising that no Smc1-Smc3-Scc2 crosslinked species can be detected. In contrast, it appears that the efficiency of Smc3-Scc2 crosslinking is not reduced in the presence of BMOE (compare lanes 2 and 4). The reason why a triple crosslink can be detected for the Smc1 E1102 BPA construct (Fig. 4D, left gel, lane 4) is probably that more sample had been loaded in this lane (compare the non-crosslinked Scc2 bands).

The cysteine pair of Smc1N1192C and Smc3R1222C have been used to probe the ATP-driven head engagement in previous studies. As shown by Chapard et al., 2019, this pair of residues were selected depending on the head-engaged crystal structure of Smc3 and Smc1 in the presence of ATP (Gligoris et al., 2014; Haering et al., 2004). His study evidenced that this crosslink is ATP-dependent using Smc3 ATP binding mutant (Smc3K38I). This is further confirmed by *in vitro* crosslink experiments (Collier et al, 2020). Although we are not certain that this crosslink would prevent ATP hydrolysis and ADP release of the engaged heads, as the reviewer pointed out, crosslinking this cysteine pair surely stabilises the ATP-dependent head-engaged conformation. We take advantage of this property to determine the interaction between Scc2 and heads-engaged cohesin.

Lane 3 cannot show an Smc1-Smc3 crosslink, as the western blot is for Scc2-FLAG. Hence, no crosslink band was observed here because this lane represents the BMOE crosslink between Smc1-MYC and Smc3-HA. **This revision contains western blots for Smc1 and Smc3 showing successful head engagement in S4H and S4I.**

We have included an over-exposed western blot for Fig 4D, and a double crosslink band still cannot be recorded, hence it is not possible that we saw a double crosslink band because we loaded more samples in the fourth lane for Smc1 E1102BPA.

6. The finding that the Smc3 QQ mutation reduces crosslinking of the Smc3 E67 residue (Figure 5) to the same extent as it reduces crosslinking of two residue pairs specifically designed for probing the engaged state (Smc3 S72 - Scc2 E819 and Smc3 K1004 - Scc2 D369) furthermore supports the conclusion that the E67 residue detects the engaged and not a previously undescribed state.

We cannot draw this conclusion just from these crosslinking efficiency data. As mentioned above, our results actually oppose “the conclusion that the **Q67** residue detects the engaged and not a previously undescribed state”. We argue that the QQ mutant inhibits the Scc2/pre-E-cohesin conformation, which prevents the transition from Scc2/J-cohesin to Scc2/E-cohesin. Therefore, Scc2/E-cohesin is also inhibited by the QQ mutant.

7. The authors claim that locking the Smc1-Smc3 heads in the engaged state by Walker B mutations increases the crosslinking efficiency of Smc3 E67 to a lesser degree than it does for two combinations designed to probe the engaged state (Figure 7A). In the absence of any quantitation of the data presented in this figure, this conclusion is hardly justified.

We mentioned that all quantification was done by taking the mean+SD of 3 independent experiments in the figure legends. We have submitted all the raw data for the quantifications for all our experiments with the revision. Additionally, we have shown all the data points in the figures in this revision.

In general, the authors should provide quantitative data for multiple repeats of each crosslinking experiment instead of just presenting a single western blot. When the authors present quantitative data, it is not clear how they corrected for different blotting efficiencies of non-crosslinked and crosslinked species. Comparing the decrease in the non-crosslinked species would probably be the most accurate way to compare different crosslinking reactions. Such comparisons are essential, for example for Figure 2C, where the non-crosslinked band of Smc3 K38I appears significantly reduced when compared to the non-crosslinked band of wild-type Smc3.

We thank the reviewer for explaining the expected quantification process, and we are happy to inform you that we followed the same way to quantify it. We have added this to the materials and methodology and added all the data points as an Excel sheet with the revision. We have also changed the graphs, showing all the data points for all our quantification.

8. The authors suggest that the interaction between Scc2 and the Smc3 coiled coil is weaker in the pre-engaged state than in the engaged state and that this interface therefore acts as an entry gate for DNA (lines 481-483). This conclusion is presumably based on the different crosslinking efficiencies of the Smc3 K1004 and Scc2 D369 residues in the Walker B mutant Smc3 (Figure 7A). Can this conclusion really be drawn from the available data? Even if distinct pre-engaged and engaged conformations existed, it is likely that the former is much more transient than the latter. Differences in crosslinking efficiencies therefore don't necessarily reflect the stability of protein-protein interactions.

We are afraid that the results mentioned by the reviewer have been misinterpreted. In this paper, we used BMOE crosslink to measure the proximity in cystine pairs which likely reflects the formation of the interface between Smc3 K1004-Scc2 D369. We showed that interaction between Smc3 K1004-Scc2 D369 residue is significantly increased by Walker B mutant, which suggests ATP-dependent head engagement brings Smc3CC near to Scc2. Because we believe Smc3 Q67-Scc2 T1175 represents pre-E conformation in which the head is disengaged, therefore, we inferred that the formation of the Scc2-Smc3CC interface is largely reduced in pre-E conformation. This is confirmed by our new double crosslink result (Fig 7C). However, Scc2 interacts with the Smc3 head in both pre-E and E conformation. Therefore, we proposed that the transient Scc2/Smc3CC interaction in the Scc2/pre-E-cohesin will be an ideal candidate entry gate for DNA clamping. Moreover, the crosslink efficiency of Smc3 Q67C/Scc2 T1175C is higher than Smc3 K1004C/Scc2 D369C, suggesting that the Scc2/pre-E-cohesin probably predominates Scc2/E-cohesin. It is not surprising because the Scc2/E-cohesin/TAP

configuration triggers ATP hydrolysis and subsequent complex disassembly. This will make this configuration more transient. To stabilise this conformation in vitro, a non-degradable ATP analogue was used in the Cryo-EM study. Otherwise, only the Scc2/J-cohesin complex was detected (Petela, 2021)

9. Several conclusions rely on the interpretation of yeast strains with four or more mutations, for example smc3 K112Q K113Q R1008I scc2 E822K L937F (Figure 6F-I). Drawing specific mechanistic conclusions from such mutant combinations needs to be done with particular care. For example, the authors first suggest that the Smc3 R1008I mutation suppresses the Smc3 QQ phenotype by stabilizing the Scc2-Smc3 coiled coil interaction (lines 350-351) but later suggest that this mutation instead destabilizes juxtaposed coiled coils (lines 554-556). It remains also unclear how the W483R mutation located close to the hinge domain of Smc3 restores the (pre-)engaged state in the Smc3 QQ mutant. The authors speculate that this mutation might, like R1008I, destabilize the juxtaposed coiled-coil conformation. This suggestion could be directly tested with the authors' crosslinking assay.

We meant that in Smc3QQ, the rescue mutant somehow changes the orientation of coiled coils in J-conformation, allowing Smc3 CC to interact with Scc2. These two statements are not contradictory; the later event (Smc3 CC's interaction with Scc2) in QQ mutant is because of the first (opening of juxtaposed coiled coil by suppressor mutants). However, further study will be needed to understand coiled-coils role in Smc3-Scc2 crosslink in WT and QQ mutants, which is out of the scope of this study.

10. In their interpretation of the experiments with acetylation-mimicking Smc3 QQ mutants, the authors ignore a potential function of the Pds5 protein, which is known to bind to these residues (Petela et al., 2021) and enacts its downstream effects on cohesion establishing (Hartman et al., 2000) and Wapl-mediated unloading (Shintomi & Hirano, 2009). Pds5 and Scc2 bind cohesin in a mutually exclusive manner, and a lack of Smc3 Q67 crosslinking to Scc2 is an expected result of Scc2 replacement by Pds5. Indeed, the reduced detection of Smc1-Scc2 binding in the immunoprecipitated complexes (Figure 5J) are not explained by the pre-engaged, engaged, and juxtaposed conformations. As Pds5 relies on a folded elbow to bridge the acetylated patch on the Smc3 ATPase head and the hinge domain (Petela et al., 2021), mutations that are expected to destabilize elbow folding, such as Smc3(W483R) could also reduce Pds5 binding. The authors should probe for Pds5 in their immunoprecipitation experiments (Figure 5J) and

repeat the BMOE crosslinking experiments of Smc3 QQ mutants (Figures 5A, F, H and 6A, C) with purified cohesin *in vitro* to exclude secondary effects of interference of Pds5 *in vivo*. *In vitro* experiments also enable the use of non-hydrolysable ATP analogues as an alternative to sequential BMOE and BPA crosslinking (Figure 4D).

This reviewer raised an interesting possibility that Pds5 might preferentially bind smc3QQ and inhibit Scc2/cohesin interaction. However, our co-IP results demonstrated that the global interaction of Scc2/cohesin is not affected by Smc3QQ (Fig 2E). So, it is unlikely that Pds5 affected the Smc3QQ Q67C/Scc2 T1175C crosslink. We confirmed this using an Auxin-inducible degron to deplete Pds5. The degradation of Pds5 does not influence the crosslinking efficiency between Smc3QQ Q67C/Scc2 T1175C. This demonstrated that the defective crosslink of Scc2 with the Smc3 head caused by smc3QQ is not due to Pds5. We included this result in Fig S5C.

Considering that Smc3 Q67 crosslinks to Scc2 are compatible with the structure reported for DNA-bound engaged cohesin (Collier et al., 2020), a simpler interpretation of the data in Figure 4 is that DNA-binding of the mutant is much reduced (Figure 1C) and hence fewer complexes can be found in the DNA-bound engaged conformation, either probed for by Smc3 Q67 (Figure 4B), S72 (Figure 4G), or K1004 (Figure 4I).

We are characterising the Scc2/pre-E-cohesin complex *in vitro*. Our unpublished *in vitro* smc3 Q67C/Scc2 T1175C crosslink is neither DNA- nor ATP-dependent, which excluded this possibility.

Minor comments:

11. The abstract should mention that the study was performed in yeast.

Thanks to the reviewer for pointing this out and we have included that information in the abstract.

10. Line 150: To compare ATP hydrolysis rates in the presence of Scc2 and DNA, the authors should determine hydrolysis rates (ATP molecules hydrolysed per second per cohesin molecule) from the linear part of the curves shown in Figures 2A and B and determine average rates from independent repeats of the experiment. The same comment applies to Figures 6 E, H, and I.

This assay and the data presentation have been accepted in previous studies (Petela 2018). The key information we try to convey in this study is the difference between ATP hydrolysis activity between WT and QQ, which is clearly shown by our data.

12. Line 186: Q66 should be H66.

Thanks to the reviewer for pointing this out. H67 has been changed to H66

13. Lines 515-516: “reversing the positive charge of Scc2 E822 leads to a gain of-function allele of Scc2 (scc2-E822K)”. The authors probably wanted to write that the mutation reversed the negative charge of glutamate.

Thank you, reviewer, for pointing this out. We have re-formatted the sentence for a clear understanding.

13. All figures with Western blots: it would be extremely helpful if the authors indicated size markers on each blot to allow the reader to compare the assignment of crosslinked species. Some of the gels have been extensively cropped (e.g., Figure 2E).

We have indicated the molecular marker points to our main figures with the revision and all crosslinked products have been verified by counter antibody check. E.g., for each Smc3-HA-Scc2-FLAG crosslink, we have presented both anti-HA and anti-FLAG western blots. Also, We have submitted all the original western blot with the revision in source data.

14. Figure 1C: Blue and green curves almost indistinguishable.

We thank the reviewer for pointing this out. We have changed one of them to purple.

15. Figure 1D: W483R is mislabelled as W384R.

We thank the reviewer for pointing this out. We have amended the mistake.

16. Figure 1E: 483R should be W483R.

We thank the reviewer for pointing this out. We have amended the mistake.

17. Figure 1G: A smc3QQ delta-eco1 control strain should be included in the experiment (which should be viable).

This is impossible as smc3QQ won't be loaded to DNA in the first place.

18. Figure 4B: M74C is mislabelled as M47C.

We thank the reviewer for pointing this out. We have amended this.

19. Language editing will be required throughout the manuscript (for example, lines 478-479: “the heads in the Scc2/pre-E-cohesin complex are disengaged, which is unable to bind ATP.”; lines 522-524: “so smc3QQ is still able to form a complex with Scc2 interaction, which is expected in the Scc2/J-cohesin configuration”; lines 933-935: “After immunoprecipitated against Scc1-PK and separated on gradient gel, Scc2...”).

Thanks to the reviewer for the suggestion and we have done major reformatting of the paper.

20. Even if the manuscript addresses a specialized audience, the authors nevertheless need to outline their experimental strategies. For example, the crosslinking experiment shown in Figure 1H is impossible to interpret without some additional explanation in the figure legend. In general, additional information could be provided in the figure legends.

Thanks to the reviewer for the suggestion, and we have added more information in the legends, throughout the paper.

21. The authors need to carefully check their methods section. For example, were yeast cells really incubated shaking at 2000 rpm (line 585)?

Thanks to the reviewer for pointing this out and we have corrected it to 200rpm.

22. Both supplementary tables are labelled as S1.

Thanks to the reviewer for pointing this out and we have now labelled the second table as S2.

Reviewer #3 (Remarks to the Author):

The manuscript by Kaushik et al., extends our knowledge of the molecular mechanism by which the chromosome-organising complex, cohesin functions. The authors use elegant and sophisticated cross-linking experiments in budding yeast together with structural predictions to provide evidence for the existence of cohesin-Scc2 configurations in vivo. These configurations include two configurations previously identified through in vitro structural studies (Scc2 in complex with E-cohesin and J-cohesin), providing crucial evidence that these conformations exist also in vivo. The authors additionally identify a new configuration, which they name “pre-E-cohesin”, which is likely a

key intermediate in the structural transition between E- and J-cohesin. The authors go on to show, using an Smc3 acetylation mimicking mutation (smc3-QQ), that Smc3 acetylation prevents Scc2 binding to E-cohesin and pre-E cohesin. Using a genetic screening approach, the authors additionally identify mutations in SMC3 that overcome the lethality of the Smc3-QQ. They show that these smc3-QQ-suppressing mutations enhance Scc2 binding to E-cohesin and pre-E cohesin, allowing them to link the structural transitions of cohesin to cellular function (cohesin levels on chromosomes). This is a high quality, extremely thorough and insightful study with rigorously performed experiments and important conclusions. The findings will be of great interest to the chromosome biology and molecular motor fields. The experiments and data are beautiful. However, the manuscript needs significant re-writing to make the findings accessible and to ensure that the study has the impact it deserves.

Major concerns

1. The manuscript is very dense and challenging to decipher. Please re-write with clear explanations for the non-specialist reader. There are many places where more explanation is needed or where sentences do not make sense. The most prominent are listed in “minor concerns”.

Thanks to the reviewer for the suggestion and we have done major reformatting of the paper.

2. Figure legends are sparse and lack key information. Please provide more details of the experiments (which antibody?, are the blots from whole cell extracts etc.) to allow the reader to fully understand the experiments performed.

Thanks to the reviewer for the suggestion, and we have added more information in the legends, throughout the paper.

3. On many figures the antibody used for western blotting is not labelled which makes the figure difficult to interpret. Please ensure that it is clear how and what is being detected on the blots.

Thanks to the reviewer for the suggestion, and we have added more information in the figures and legends, regarding western blots, throughout the paper.

4. The rationale for the screen in which smc3E199A and smc3R10081 were identified is not clear. These mutations were identified as suppressors of the wpl1 Δ and scc2-45 synthetic lethality. First, it is not clear why the absence of the Wpl1 cohesin-destabilising protein would be synthetic lethal with scc2-45 where less cohesin should be loaded. Second, it is not clear why such a

suppressive mutation would be expected to rescue smc3-QQ. Is the rationale that wpl1Δ scc2-45 is not viable because scc2-45 relies on cohesin recycling by Wpl1 to have sufficient cohesin for cohesion? And that smc3-QQ would similarly reduce the availability of cohesin for loading? Please provide an explanation for the logic behind this screen and its application to smc3-QQ.

The origin of these two suppressors is not related to this study. To make the paper more accessible, we removed the description of how we got these two suppressors. The synthetic lethality of wpl1Δ and Scc2 ts mutant was discovered many years ago (Curr Biol, 2009, 19:492) and confirmed by our unpublished experiment. We found that Wpl1 plays a role in loading and might contribute to Scc2/cohesin complex formation. This will be a different story.

5. The experiments presented in Figure 5 make the assumption that the interaction between Scc2-Smc3 CC is a marker of engaged heads (E-cohesin). However, the evidence that shows that this is the case is only presented in Figure 7. A more logical presentation of the data would therefore be to present the data in Figure 7 before current Figure 5.

Thanks to the reviewer for the suggestion, but we beg to differ in the order of data presentation. Figure 5 focuses on the effects of the QQ mutant on different configurations of the Scc2/cohesin complexes, which directly echoes Fig 1 and 2. The reported cryo-EM data revealed Smc3CC/Scc2 interface in E-cohesin, which is a good rationale to do the experiment. The finding about the role of ATP binding on Smc3 CC/Scc2 interaction is another key discovery in this paper, which was discussed after the story of Smc3QQ. We believe that this arrangement would make a smoother logical flow of this paper.

6. The authors use their elegant cross-linking approach to probe specific interactions of Scc2 with Smc3QQ and conclude by a process of elimination that J-cohesin can still bind. However, they cannot rule out that Smc3-QQ could bind distinct cohesin conformations that are yet to be discovered. Though not essential, it would strengthen the manuscript to determine whether it Scc2 can really associate with Smc3-QQ J-cohesin. Could the authors lock Smc3-QQ cohesin in the J- or E conformation (as in Figure 4) to address this point?

We thank the reviewer for the suggestion, and we agree with them. However, the experiment is extremely technically challenging since smc3QQ has to be ectopically expressed and its associated

Scs2/J-cohesin needs to be separated from endogenous Smc3, which greatly reduces the detection sensitivity of double crosslink products. Therefore, the result might be misinterpreted.

7. Throughout the manuscript, including the introduction and discussion, the emphasis is on cohesin loading. However, Scs2 is also driving loop extrusion and this should also be discussed in the context of what they found. Only the rescue of viability can be specifically connected to loading.

Thanks to the reviewer for the suggestion and we have discussed our model considering the loop extrusion as well.

8. The ATPase assays in Figure 6H and I: are they with DNA or not? This information needs to be included in the figure and legend. If there is DNA why is the result different from the wild type data in Figure 2A. If there is DNA, why does the wild type show such a strong response in Figure 6H but not 2A? If there is DNA, why doesn't the QQ mutant show a stronger response in Figure 6I.

Thanks to the reviewer for pointing this out. They are not with DNA; we have included that in the figure legend.

Minor comments

1. Figure S1A. It would be useful to include this in the main figures for ease of understanding.

Thanks to the reviewer for pointing this out. We have changed the position.

2. Page 2, line 46. This sentence is difficult to understanding and too simplistic. The authors should discuss the findings from Bauer 2021 in their manuscript, in particular the knowledge that cohesin has more than 2 configurations and that it undergoes cyclical structural changes.

We agreed with this reviewer that there are more configurations of Scs2/cohesin during loop extrusion as described in Bauer 2021, which has been included in the introduction. However, this paper is to study the loading process, which leads to a topological entrapment. The other non-topological interaction of Scs2/cohesin found in loop extrusion might not reflect the loading reaction. But the reviewer raised this interesting point, which lead to a further discussion on how the conformational dynamics revealed by this study involve loop extrusion. We have included this in the discussion.

3. Page 3, line 82. "corroborating" does not fit here. Perhaps the authors could simply say that

they identified a new configuration in addition to the two published ones? However, see point 2 above.

Thanks to the reviewer for pointing this out, and we made the change accordingly.

4. line 85. Should read “the DNA clamp”. However this is the first mention and the DNA clamp has not been described. Some introduction/explanation is needed. As from the points above, a more general and complete description of cohesin structural transitions/models is required.

We change to “DNA clamping” and added more introduction to DNA clamping in the introduction. However, due to the word limit, we cannot provide more detailed information.

5. Line 104. “evolved” suggests an evolution experiment, while the authors actually carried out a random mutagenesis experiment. The authors could simply state that they aimed to identify intragenic suppressors by random mutagenesis.

Thanks to the reviewer for pointing this out, and we reframed the statement.

6. Line 105. Where were the other mutations?

The rest mutations are located on the head, as described in Fig S6G

7. Line 131. This is the first mention of BMOE. Please explain the approach.

Thanks to the reviewer for the suggestion and we have added the information.

8. Line 144 and throughout, please correct nomenclature. Genes in italics (capitals for wild type and lowercase for mutations), proteins in title case non-italic. There are many instances in the figures and text where this is incorrect.

Thanks to the reviewer for pointing this out, and we have corrected the nomenclature.

9. Line 182 “KKD strand”. It is not immediately obvious what this means or that this is the residues that are acetylated by Eco1. More explanation is required. It is also not necessary to abbreviate unless it will be used again later.

Thanks to the reviewer for pointing this out, and we have added more information about KKD.

10. Line 184. Could the figure be improved to make the potential interaction surfaces easier to understand?

As it is a crystal structure, we do not have much freedom in placing and labelling the residues, which does not help to understand the residue position without zooming in. We tried a few times, and for ease of understanding, we included the crystal structure used for deciding these residues in the Source Data.

11. Figure 3B. The inability to detect the interaction does not mean that they do not interact - perhaps the substitutions are not appropriately positioned. The structure does show the interaction so this interaction seems likely. The interpretation of this experiment should be adjusted to account for this.

Thanks to the reviewer for pointing this out, and we agree with this point. e.g., Smc3 L111BPA might mimic QQ effects as it is a neighbouring residue, thereby reducing the crosslink efficiency. We have now added these points to our manuscript.

12. Figure S3A and throughout. Figure and figure legends need to state which antibodies used for western blot. More details of the assay are required. Is this whole cell extract or immunoprecipitated protein?

Thanks to the reviewer for the suggestion, and we have added more information in the legends, regarding western blots, throughout the paper.

13. Line 209. Summary sentence is not very informative. It would be useful for the reader to state what the two interfaces are.

Thanks to the reviewer for the suggestion, and we have reformatted the sentence.

14. Figure S3D. This is an elegant experiment but the figure is laid out in a very confusing way. Flag is coloured in green but on the blot, the green dots refer to the lack of Flag. Also the westerns are flipped so that the HA blot is aligned with schematic which includes Flag, it would be better to reverse them. Also label the blot with antibody used – this is critical to understand the experiment.

Thanks to the reviewer for the suggestion, and we have added more information to the western blot. We have also changed the colour for the dots and arranged the western blot so that it matches to the schematic.

15. Figure 3H. It would be helpful to indicate the residues used to generate cysteine pairs in a close up of this model.

Thanks to the reviewer for pointing this out. We have replaced the figure with a more closed up version and we have now indicated the residues. Additionally, we have submitted the model with this revision.

16. Line 249. Not really a neighbour residue. Could the authors show this is Figure S4A/B?

We have now indicated Smc M74, its respective Scc2 crosslink partner residues and Smc3 Q67 for E and pre-E in S4C and S4D, in this revision.

17. Line 259. "predict a fact" is awkward. " "Makes two predictions" would be better.

Thanks to the reviewer for the suggestion, and we have formatted the sentence.

18. Line 260. The phrasing here is difficult to understand. The authors should say that the model predicts that the CC are separate.

Thanks to the reviewer for the suggestion, and we have formatted the sentence.

19. Figure 4C. Please label on the plot that this is a three protein interaction for ease of understanding. Smc3-Scc2-Smc1. It is not correct to call it J-cohesin as this is not what is on the blot after denaturing.

Thanks to the reviewer for the suggestion, and we have formatted the figure label.

20. Line 295. Please indicate in the figure that this is Scc2-T1175C.

Thanks to the reviewer for the suggestion, and we have formatted the figure label.

21. Figure 5C. It is difficult to discern Scc2 and Smc3 in this figure as the colours are very similar, although it is appreciated to use the same colour scheme throughout the manuscript. Consider changing the colours. The schematics showing positions of cross-links are extremely useful.

We have changed the color for Scc2.

22. Line 334. "This conclusion was supported.. (data not shown)." This sentence does not provide any additional clarity/confirmation without better explanation. Either delete the sentence or show the data.

Thanks to the reviewer for the suggestion, and we have deleted the sentence.

23. Line 336. The conclusion that Scc2 interacts with this interface in the J configuration is too

strong because there may be other configurations that we don't know about. It also cannot be ruled out that Scc2 interactions with 1102 in the E or pre-E configurations but less efficiently. Although not required, this could be tested by locking cohesin in the J configuration and then testing binding.

Thanks to the reviewer for the suggestion, and we tuned down our statement on this point and included an alternative explanation in the paper. As our reply to reviewer 1 point 6, the proposed experiment is extremely technically challenging, and the result might be misinterpreted.

24. Figure 6C. DO the authors have a stronger exposure of this blot? It is difficult to see the bands referred to.

Here we are presenting an over exposed blot. (Anti-HA)

25. Figure 6E. The authors should show wild type and smc3-QQ controls for comparison.

The mentioned data has been added to fig 6F.

26. Line 362. The summary is confusing. Do the authors mean to provide an explanation for why loading is rescued, even though Scc2 only mildly stimulates ATPase activity? Is this explanation the idea that because Scc2 binds more efficiently to Smc3QQ this brings the complex to the DNA, allowing further stimulation of ATPase?

Yes, the idea raised by this reviewer is what we try to convey. We rephrased these descriptions to clarify our explanation

27. Please call out figures in order in the text.

Thanks to the reviewer for the suggestion.

28. Figure 6 legend, letters are mixed up.

Thanks to the reviewer for pointing this out, and we have amended this.

29. In Figure 7, are the authors using the EQ mutant for both Smc1 and Smc3? If so, this should be indicated in the figure.

Thanks to the reviewer for pointing this out. We are using Smc3EQ for Smc3-Scc2 interface and Smc1EQ for Smc1-Scc2 interface. We have added this information to the figures and legends

30. The discussion should be shortened as the main points are lost. The authors are encouraged to provide concise discussion focused on three points: 1. The pre-E state they identified. 2. Confirmation of in vivo existence of configurations predicted from structural studies. 3, How acetylation affects Scc2 interaction with cohesin.

Thanks to the reviewer for the suggestion, and we have redrafted the discussion section.

31. In the methods “collect 30 OD of cells” does not make sense. What volume?

The volume depends on the OD/ml of cells. Depending on that, we centrifuged the required volume of culture for 30OD. E.g. from a culture of 0.3OD/ml, we took $30 \div 0.3OD = 100$ ml culture and centrifuged.

REVIEWER COMMENTS

Reviewer #1 (Remarks to the Author):

The manuscript has been improved following the suggestions from the reviewers. The main conclusion that Smc3 acetylation and coiled coil dynamics together control the cohesin ATPase cycle for DNA loading (and possibly also loop extrusion) is well supported.

The manuscript is still somewhat hard to follow and comprehend which is at least partly due to the many mutations given in the text, legends, and figures. The authors should consider reducing the listing of residue mutations by naming the cysteine reporters and providing an overview figure or table for all the point mutations and their usage.

Figure 1: uniform labeling of ChIP-Seq panels D and G

Figure 4, CD: typo 'stabilised'

Figure 7B: alignment of lanes and labels is not good

Listing of cysteine mutations is hard to follow. Give a name to cysteine pairs as well as an overview figure/table.

Line 143: confusing sentence. do the authors mean probing for Smc1N1192C (by immunoblotting)?

Misleading headline: 'smc3QQ barely affects head engagement' This is true when head engagement is considered to include the E-cohesin state as well as the pre-E state. Reword to avoid confusion?

Consistency in labeling Smc1 N1192C or Smc1N1192C?

Incomplete sentence: 'Cohesin loader Scc2/4 complex is a, essential for its various functions, such as sister chromatid cohesion and loop extrusion³⁴⁻³⁶.'

Reviewer #3 (Remarks to the Author):

While the data in this manuscript is strong and the discoveries are important and interesting, the authors have only partially addressed the concerns of the reviewers. The manuscript is still very difficult to read and only slightly improved compared to the previous version. There are also many typographical and grammatical errors which careful proof-reading would circumvent. The authors could also have taken more effort to address the comments raised in the previous version of the reviews and more care in presentation.

Points that were not addressed from the last review

1. Although some restructuring has helped, the manuscript is still very difficult to understand. The authors are encouraged to consider the non-specialist reader and avoid the use of acronyms as much as possible and provide context to explain which surfaces the cross-linked residues connect. It is very difficult for the reader to remember which state is being tested as they read through the manuscript. (previous point 1).

2. The authors have labelled some of the figures with the key information, but not all and not in the supplementary. However, it is important that all figures can be easily understood, including the supplementary (previous points 2 & 3).

3. Regarding previous point 5, the rationale given about the assumption that the Scc2-Smc3 CC is a marker of engaged heads is not entirely convincing. How is it known already that interaction at Smc3-S72C/Scc2-E819C and Smc3-K1004C/Scc2-D369C is specific for E cohesin? Seeing such interaction on the structure is not sufficient to conclude that it happens in vivo, nor to conclude that it's specific for E cohesin. Figure 5DE show that this interaction can happen in vivo, but which conformation allows this interaction. Therefore, from figure 5F-I it can only be concluded that residues Smc3-S72C/Scc2-E819C and Smc3-K1004C/Scc2-D369C do not interact in the QQ mutant. Evidence that such interactions are specific for E-cohesin only come from Figure 7, when it's shown that the interaction is present only when ATP is bound, aka in the engaged configuration. If the authors want to conclude that the QQ mutant affects the E configuration from figure 5, then they need to show figure 7 first. Otherwise, that conclusion can only be drawn after figure 7.

4. Previous point 8. Please include the statement in the text that the ATPase assay is not with DNA. The authors did not address the point about why Fig 6H looks so different from the non-DNA curves in Figure 2AB. This needs to be explained.

5. Previous point 24. Why not include this more exposed blot in the paper?

6. Previous point 25. "Figure 6E. The authors should show wild type and smc3-QQ controls for comparison. The mentioned data has been added to fig 6F" Figure 6F is a tetrad dissection and the requested data does not appear to have been added.

7. Previous point 27. " In Figure 7, are the authors using the EQ mutant for both Smc1 and Smc3? If so, this should be indicated in the figure. Thanks to the reviewer for pointing this out. We are using Smc3EQ for Smc3-Scc2 interface and Smc1EQ for Smc1-Scc2 interface. We have added this information to the figures and legends." Is one of these mutations sufficient to lock cohesin in the E conformation?

Further/specific points

1. Supplementary figure legends need a lot more detail. The set-up of the experiment is always missing but needs to be described.

2. The first time that the C-C X-link experiment is presented (Fig 2C) it should be better explained how it is set up: is it in vivo? In which cell cycle stage? Is it from whole cell extracts or is there some purification?

3. If the QQ mutant and the QQ WR mutant are not subject to Wpl1 mediated release, while wild type cohesin is, then any difference in cohesin levels found by ChIP cannot be solely attributed to a difference in loading. Therefore, from Fig 1D and 1G it can only be concluded whether QQ and QQ WR can be loaded on DNA or not, comparisons with wild type are confounded. Sentences at lines 98, 289-290, 309, 331 need to be changed.

4. Figure S4E,H,I are never called, nor described in the main text. Moreover, the description in the figure legend is not sufficient to understand what they are showing.

5. Please explain figure S4G. Sentence in the text (line 234-6) doesn't make sense without further explanation.

6. The paragraph from line 340 onward is extremely difficult to follow. Please help the reader by saying clearly every time which surface in which conformation is being tested.

7. The definition of engaged head conformation comes from structures of the EQ mutant. If the Q67 crosslink is specific for preE, why is it present also in the EQ mutant? Is it because only Smc3 is mutated and not Smc1? Does that mean that heads are not actually engaged in the experiment in figure 7A? The same question could be asked for the other crosslink indicative of J cohesin. (lines 344-347)

8. The use of CC for coiled-coil is very confusing. It can be easily mixed with a conformation (like E or J) or with the C-C crosslink. It would be better to write it out in full.

9. Several western blots are still missing details in the figures, as are the two ChIP experiments.

10. Gene/protein/mutant nomenclature is still incorrect in multiple cases

11. Text is often incomprehensible and full of typos. Very careful proof-reading is required throughout. Examples below (not an exhaustive list).

- Line 370, "Scc2/4 complex is a, essential.." There is something missing here.
- Line 406: pre-E-cohesin

12. Many figures are mis-called. This needs to be carefully checked throughout. Examples are: -
Line 269 Should be S5E, not C

- Line 270 should be S5F and G, not E and F
- Line 296 should be 6D and E, not S6D and E. Also, explain in the text what interface is represented by those residues. Smc3 head/Scc2?
- Line 300: not 6D. And please rephrase to call figures in order.
- Line 306 should be 2A-B, not 1F
- What's the difference between Fig 6E and S6F? it's not explained anywhere.
- Line 316 should be S6G not F
- Line 336 should be S6I, not H
- Should be Fig. 1H

This document contains reviewers' comments in black and authors' responses in red.

REVIEWER COMMENTS

Reviewer #1 (Remarks to the Author):

Figure 1: uniform labeling of ChIP-Seq panels D and G.

We thank the reviewer for pointing this out, and we have uniformised the labelling.

Figure 4, CD: typo 'stablised'

We thank the reviewer for pointing this out, and we have corrected the spelling.

Figure 7B: alignment of lanes and labels is not good

We thank the reviewer for pointing this out. We have realigned the labels with the lanes and also shifted figure 7B to Figure 5E.

Listing of cysteine mutations is hard to follow. Give a name to cysteine pairs as well as an overview figure/table.

We clarified the interfaces and the configurations of the cysteine reporters in the main text. We also provided an overview table for the mutations as Supplemental table 1.

Line 143: confusing sentence. do the authors mean probing for Smc1N1192C (by immuno-blotting)?

Thank the reviewer for pointing this out. We changed this statement to "A similar conclusion was drawn by examining the Smc1^{N1192C} using anti-Myc antibody".

Misleading headline: 'smc3QQ barely affects head engagement' This is true when head engagement is considered to include the E-cohesin state as well as the pre-E state. Rerword to avoid confusion?

"Head engagement" in this paper was defined as "ATP-mediated head engagement", which forms ATP-binding pockets and binds ATP. To avoid confusion, we revised it to "ATP-mediated head engagement".

Consistency in labeling Smc1 N1192C or Smc1N1192C?

We have corrected the Gene/protein/mutant nomenclature as the following:

Gene: Wild type- *SMC3*; mutants- *smc3-QQ* or *smc3-QQ_W483R_R1008I*

Protein: Wild type- Smc3; mutants- Smc3^{QQ} or Smc3^{QQ_W483R_R1008I}

Residues: Wild type- Smc1 N1192; mutations- *smc1 N1192C*

Incomplete sentence: 'Cohesin loader Scc2/4 complex is a, essential for its various functions, such as sister chromatid cohesion and loop extrusion34–36.'

Thank you for pointing it out. We removed "a,"

Reviewer #3 (Remarks to the Author):

1. Although some restructuring has helped, the manuscript is still very difficult to understand. The authors are encouraged to consider the non-specialist reader and avoid the use of acronyms as much as possible and provide context to explain which surfaces the cross-linked residues connect. It is very difficult for the reader to remember which state is being tested as they read through the manuscript. (previous point 1).

Thank you for pointing it out. We realized that so many mutations and cysteine pairs would cause considerable confusion to readers. We, therefore, stated the interfaces and the configurations represented by the cysteine pairs in the main text and included the acronyms in parentheses. We also provided an overview table for the mutations as Supplemental table 1.

2. The authors have labelled some of the figures with the key information, but not all and not in the supplementary. However, it is important that all figures can be easily understood, including the supplementary (previous points 2 & 3).

We thank the reviewer for their suggestion and patience. We have now included more details in the supplementary figures.

3. Regarding previous point 5, the rationale given about the assumption that the Scc2-Smc3 CC is a marker of engaged heads is not entirely convincing. How is it known already that interaction at Smc3-S72C/Scc2-E819C and Smc3-K1004C/Scc2-D369C is specific for E cohesin? Seeing such interaction on the structure is not sufficient to conclude that it happens in vivo, nor to conclude that it's specific for E cohesin. Figure 5DE show that this interaction can happen in vivo, but which conformation allows this interaction. Therefore, from figure 5F-I it can only be concluded that residues Smc3-S72C/Scc2-E819C and Smc3-K1004C/Scc2-D369C do not interact in the QQ mutant. Evidence that such interactions are specific for E-cohesin only come from Figure 7, when it's shown that the interaction is present only when ATP is bound, aka in the engaged configuration. If the authors want to conclude that the QQ mutant affects the E configuration from figure 5, then they need to show figure 7 first. Otherwise, that conclusion can only be drawn after figure 7.

We now understand the valid point raised by this reviewer. We changed the order of the figures and moved Fig 7 to Fig 5.

4. Previous point 8. Please include the statement in the text that the ATPase assay is not with DNA. The authors did not address the point about why Fig 6H looks so different from the non-DNA curves in Figure 2AB. This needs to be explained.

We have clarified that the ATPase assay was performed in the absence of DNA. The difference between Fig 7HI and Figure 2AB is due to different patches of recombinant proteins. The recombinant proteins used in Fig 7HI were prepared at the same time and, somehow, showed higher ATPase activities. This situation is not rare in our hands. However, the key is that the expression/purification of proteins and ATPase assay in one experiment have to be performed side by side, which is what we did for Fig 2AB or 7HI.

5. Previous point 24. Why not include this more exposed blot in the paper?

We thank the reviewer for their suggestion and added this figure as Fig 7C.

6. Previous point 25. "Figure 6E. The authors should show wild type and smc3-QQ controls for comparison. The mentioned data has been added to fig 6F" Figure 6F is a tetrad dissection and the requested data does not appear to have been added.

We thank the reviewer for pointing out the mistake. And we agreed with this reviewer that a direct comparison of WT/QQ/QQ_WR_RI's ATPase activities would give a clearer view of the effect of WR_RI on QQ's ATPase activity. To this end, we replaced this figure with a new set of data (Fig 7E)

7. Previous point 27. "In Figure 7, are the authors using the EQ mutant for both Smc1 and Smc3? If so, this should be indicated in the figure. Thanks to the reviewer for pointing this out. We are using Smc3EQ for Smc3-Scc2 interface and Smc1EQ for Smc1-Scc2 interface. We have added this information to the figures and legends." Is one of these mutations sufficient to lock cohesin in the E conformation?

The evidence so far for cohesin EQ mutants to improve head engagement comes from our previous *in vitro* experiment (Fig 5, Current Biology, 2012, 21:12-24). As shown in this experiment, single EQ mutations increased the engaged heads to around 30% and double EQ mutants to ~90%, compared to WT with nearly 0%. However, this interaction demonstrated by gel filtration became undetectable in the pull assay (unpublished results). This suggested that EQ mutations could not lock the engaged heads and the heads still fall into the E- to J- cycles. However, EQ mutations delayed the transition from E- to J- and prolonged the E-status, which is sufficient for E-Smc3 coiled-coil/Scc2 crosslink. Therefore, we can detect Scc2/pre-E-cohesin and Scc2/E-cohesin in EQ mutants using *in vivo* crosslink.

Further/specific points

1. Supplementary figure legends need a lot more detail. The set-up of the experiment is always missing but needs to be described.

We thank the reviewer for their suggestion and added more details to the supplementary legends.

2. The first time that the C-C X-link experiment is presented (Fig 2C) it should be better explained how it is set up: is it *in vivo*? In which cell cycle stage? Is it from whole cell extracts or is there some purification?

We thank the reviewer for their suggestion, and we have added more details to the text.

3. If the QQ mutant and the QQ WR mutant are not subject to Wpl1 mediated release, while wild type cohesin is, then any difference in cohesin levels found by ChIP cannot be solely attributed to a difference in loading. Therefore, from Fig 1D and 1G it can only be concluded whether QQ and QQ WR can be loaded on DNA or not, comparisons with wild type are confounded. Sentences at lines 98, 289-290, 309, 331 need to be changed.

We agree with the reviewer that the direct comparison of DNA association of WT with smc3-QQ-WR might be misleading. We therefore revised the statement as "WR greatly improved the loading of smc3-QQ".

4. Figure S4E,H,I are never called, nor described in the main text. Moreover, the description in the figure legend is not sufficient to understand what they are showing.

We thank the reviewer for pointing this out. We have added to the figure reference in the text and added information to the figure legends.

5. Please explain figure S4G. Sentence in the text (line 234-6) doesn't make sense without further explanation.

We included a more detailed explanation of this figure in the main text, which stated that the extensive contact between juxtaposed coiled coils was confirmed by these cysteine crosslinks and the pair with the highest efficiency was used to fix J-cohesin.

6. The paragraph from line 340 onward is extremely difficult to follow. Please help the reader by saying clearly every time which surface in which conformation is being tested.

We clarified these interfaces/configurations in the main text.

7. The definition of engaged head conformation comes from structures of the EQ mutant. If the Q67 crosslink is specific for preE, why is it present also in the EQ mutant? Is it because only Smc3 is mutated and not Smc1? Does that mean that heads are not actually engaged in the experiment in figure 7A? The same question could be asked for the other crosslink indicative of J cohesin. (lines 344-347)

Please check our response to main point 7.

8. The use of CC for coiled-coil is very confusing. It can be easily mixed with a conformation (like E or J) or with the C-C crosslink. It would be better to write it out in full.

We thank the reviewer for pointing this out and we have now used the full-form coiled coil in place of CC.

9. Several western blots are still missing details in the figures, as are the two ChIP experiments.

Thanks to the reviewer for pointing this out, and we have added more details to the figures.

10. Gene/protein/mutant nomenclature is still incorrect in multiple cases

We thank the reviewer for pointing this out. We have corrected the Gene/protein/mutant nomenclature as the following:

Gene: Wild type- *SMC3*; mutants- *smc3-QQ* or *smc3-QQ_W483R_R1008I*

Protein: Wild type- Smc3; mutants- Smc3^{QQ} or Smc3^{QQ_W483R_R1008I}

Residues: Wild type- Smc1 N1192; mutations- *smc1 N1192C*

11. Text is often incomprehensible and full of typos. Very careful proof-reading is required throughout. Examples below (not an exhaustive list).

- Line 370, "Scc2/4 complex is a, essential.." There is something missing here.

- Line 406: pre-E-cohesin

Thanks to the reviewer for pointing this out, and we have done careful proofreading to avoid these mistakes.

12. Many figures are mis-called. This needs to be carefully checked throughout. Examples are: - Line

269 Should be S5E, not C

- Line 270 should be S5F and G, not E and F

- Line 296 should be 6D and E, not S6D and E. Also, explain in the text what interface is represented by those residues. Smc3 head/Scs2?

- Line 300: not 6D. And please rephrase to call figures in order.

- Line 306 should be 2A-B, not 1F

- What's the difference between Fig 6E and S6F? it's not explained anywhere.

- Line 316 should be S6G not F

- Line 336 should be S6I, not H

- Should be Fig. 1H

We thank the reviewer for their patience and for pointing this out. We have done careful proofreading and amended the figure references as required.

REVIEWERS' COMMENTS

Reviewer #3 (Remarks to the Author):

The authors have addressed the remaining concerns. There are a few remaining corrections required.

1. Line 304 In the title, "might" would be better replaced by "potentially"
2. Line 344-345. There is something wrong with this sentence, did the authors mean to insert "because" after "It is not unexpected"?
3. Line 403-403 Typo: It should read "Scs2-Smc1 head interface".
4. Table S1 has tracked changes
5. Please provide titles for the supplementary figures

This document contains reviewers' comments in black and authors' responses in red.

REVIEWER COMMENTS

Reviewer #3 (Remarks to the Author):

The authors have addressed the remaining concerns. There are a few remaining corrections required.

1. Line 304 In the title, "might" would be better replaced by "potentially"

We followed the suggestion from this reviewer to replace "might" with "potentially".

2. Line 344-345. There is something wrong with this sentence, did the authors mean to insert "because" after "It is not unexpected"?

Thank the reviewer for pointing it out. We added "because" after "It is not unexpected".

3. Line 403-403 Typo: It should read "Scc2-Smc1 head interface".

Again, thank the reviewer for pointing it out. We corrected the typo as "Scc2-Smc1 head interface".

4. Table S1 has tracked changes

We aligned the tracks in the Table S1

5. Please provide titles for the supplementary figures

The titles of the supplementary figures are added in the supplementary files.